# ON SIZE GENERALIZATION IN GRAPH NEURAL NETWORKS

## ABSTRACT

Graph neural networks (GNNs) can process graphs of different sizes but their capacity to generalize across sizes is still not well understood. Size generalization is key to numerous GNN applications, from solving combinatorial optimization problems to learning in molecular biology. In such problems, obtaining labels and training on large graphs can be prohibitively expensive, but training on smaller graphs is possible.

This paper puts forward the size-generalization question and characterizes important aspects of that problem theoretically and empirically. We prove that even for very simple tasks, such as counting the number of nodes or edges in a graph, GNNs do not naturally generalize to graphs of larger size. Instead, their generalization performance is closely related to the distribution of local patterns of connectivity and features and how that distribution changes from small to large graphs. Specifically, we prove that for many tasks, there are weight assignments for GNNs that can perfectly solve the task on small graphs but fail on large graphs, if there is a discrepancy between their local patterns. We further demonstrate on several tasks, that training GNNs on small graphs results in solutions which do not generalize to larger graphs. We then formalize size generalization as a domain-adaption problem and describe two learning setups where size generalization can be improved. First, as a self-supervised learning problem (SSL) over the target domain of large graphs. Second as a semi-supervised learning problem when few samples are available in the target domain. We demonstrate the efficacy of these solutions on a diverse set of benchmark graph datasets.

## 1 INTRODUCTION

Graphs are a flexible representation, widely used for representing diverse data and phenomena. Graph neural networks (GNNs) – Deep models that operate over graphs – have emerged as a prominent learning model (Bruna et al., 2013; Kipf and Welling, 2016; Veličković et al., 2017). They are used in natural sciences (Gilmer et al., 2017), social network analysis (Fan et al., 2019), for solving difficult mathematical problems (Luz et al., 2020) and for approximating solutions to combinatorial optimization problems (Li et al., 2018).

In many domains, graphs data vary significantly in size. This is the case for molecular biology, where molecules – represented as graphs over atoms as nodes – span from small compounds to proteins with many thousands of nodes. It is even more severe in social networks, which can reach billions of nodes. The success of GNNs for such data stems from the fact that the same GNN model can process input graphs regardless of their size. Indeed, it has been proposed that GNNs can generalize to graphs whose size is different from what they were trained on , but it is largely unknown in what problems such generalization occurs. Empirically, several papers report good generalization performance on specific tasks (Li et al., 2018; Luz et al., 2020). Other papers, like Veličković et al. (2019), show that size generalization can fail on several simple graph algorithms, and can be improved by using task-specific training procedures and specific architectures.

Given their flexibility to operate on variable-sized graphs, A fundamental question arises about generalization in GNNs: **"When do GNNs trained on small graphs generalize to large graphs?"**

Aside from being an intriguing theoretical question, this problem has important practical implications. In many domains, it is hard to label large graphs. For instance, in combinatorial optimization

problems, labeling a large graph boils down to solving large and hard optimization problems. In other domains, it is often very hard for human raters to correctly label complex networks. One approach to this problem could have been to resize graphs into a homogeneous size. This is the strategy taken in computer vision, where it is well understood how to resize an image while keeping its content. Unfortunately, there are no effective resizing procedures for graphs. It would therefore be extremely valuable to develop techniques that can generalize from training on small graphs.

As we discuss below, a theoretical analysis of size generalization is very challenging because it depends on several different factors, including the task, the architecture, and the data. For tasks, we argue that it is important to distinguish two types of tasks, *local* and *global*. Local tasks can be solved by GNNs whose depth does not depend on the size of the input graph. For example, the task of finding a constant-size pattern. Global tasks require that the depth of the GNN grows with the size of the input graph. For example, calculating the diameter of a graph. While there are a few previous works that explore depth-dependant GNNs (e.g., Tang et al. (2020), constant depth GNNs are by far the most widely used GNN models today and are therefore the focus of this paper.

In this paper, we focus on GNNs with constant depth and study the ability of the most expressive message passing neural networks (Xu et al., 2018; Morris et al., 2019) to generalize to unseen sizes. Our key observation is that generalization to graphs of different sizes is strongly related to the distribution of patterns around nodes in the graphs of interest. These patterns, dubbed *d-patterns* (where $d$ is the radius of the local neighborhood), describe the local feature-connectivity structure around each node, as seen by message-passing neural networks and are defined in Section 3.

We study the role of *d-patterns* both empirically and theoretically. First, we theoretically show that when there is a significant discrepancy between the $d$-pattern distributions, GNNs have multiple global minima for graphs of a specific size range, out of which only a subset of models can generalize well to larger graphs. We complement our theoretical analysis with an experimental study and show that GNNs tend to converge to non-generalizing global minima, when $d$-patterns from the large graph distribution are not well-represented in the small graph distribution. Furthermore we demonstrate that the size generalization problem is accentuated in deeper GNNs.

Following these observations, in the final part of this paper, we discuss two learning setups that help improve size-generalization by formulating the learning problem as a domain adaptation problem: (1) Training the GNNs on self-supervised tasks aimed at learning the $d$-pattern distribution of both the target (large graphs) and source (small graphs) domains. We also propose a novel SSL task that addresses over-fitting of $d$-patterns. (2) A semi-supervised learning setup with a limited number of labeled examples from the target domain. The idea behind both setups is to promote convergence of GNNs to local/global minima with good size generalization properties. We show that both setups are useful in a series of experiments on synthetic and real data.

To summarize, this paper makes the following contributions. (1) We identify a size generalization problem when learning local tasks with GNNs and analyze it empirically and theoretically. (2) We link the size-generalization problem with the distribution of $d$-patterns and suggest to approach it as a domain adaptation problem (3) We empirically show how several learning setups help improve size generalization.

## 2  RELATED WORK

**Size generalization in set and graph learning.** Several papers observed successful generalization across graph sizes, but the underlying reasons were not investigated (Li et al., 2018; Maron et al., 2018; Luz et al., 2020). More recently, (Veličković et al., 2019) showed that when training GNNs to perform simple graph algorithms step by step they generalize better to graphs of different sizes. Unfortunately, such training procedures cannot be easily applied to general tasks. Knyazev et al. (2019) studied the relationship between generalization and attention mechanisms. Tang et al. (2020) observed two issues that can harm generalization: (1) There are tasks for which a constant number of layers is not sufficient. (2) Some graph learning tasks are homogeneous functions. They then suggest a new GNN architecture to deal with these issues. Our work is complementary to these works as it explores another fundamental size generalization problem, focusing on constant depth GNNs. For more details on the distinction between constant depth and variable depth tasks see Appendix A. Several works also studied size generalization and expressivity when learning set-structured inputs (Zweig and Bruna, 2020; Bueno and Hylton, 2020). On the more practical side, Joshi et al. (2019),

Joshi et al. (2020) study the combinatorial problem of traveling salesman and whether it is possible to generalize to larger sizes. Corso et al. (2020) study several multitask learning problems on graphs and evaluate how the performance changes as the size of the graphs change.

**Expressivity and generalization in graph neural networks.** (Xu et al., 2018; Morris et al., 2019) established a fundamental connection between message-passing neural networks and the Weisfeiler-Leman (WL) graph-isomorphism test. We use similar arguments to show that GNNs have enough expressive power to solve a task on a set of small graphs and to fail on it on a set of large graphs. Several works studied generalization bounds for certain classes of GNNs (Garg et al., 2020; Puny et al., 2020; Verma and Zhang, 2019), but did not discuss size-generalization. Sinha et al. (2020) proposed a benchmark for assessing the logical generalization abilities of GNNs.

## 3 THE SIZE GENERALIZATION PROBLEM

We now present the main problem discussed in the paper, that is, what determines if a GNN generalizes well to graphs of sizes not seen during training. We start with a simple motivating example showing the problem on single layer GNNs. We then show that the question of size generalization actually depends on *d-patterns*, the local patterns of connectivity and features of the graphs, and not only on their actual size.

**Setup.** We are given two distributions over graphs $P_1, P_2$ that contain small and large graphs accordingly, and a task that can be solved with 0 error for all graph sizes using a constant depth GNN. We train a GNN on a training set $\mathcal{S}$ sampled i.i.d from $P_1$ and study its performance on $P_2$.

**GNN model.** We focus on the **first order GNN** (1-GNN) architecture from Morris et al. (2019) defined in the following way:

$$\mathbf{h}_v^{(t)} = \sigma \left( W_2^{(t)} \mathbf{h}_v^{(t-1)} + \sum_{u \in \mathcal{N}(v)} W_1^{(t)} \mathbf{h}_u^{(t-1)} + \mathbf{b}^{(t)} \right).$$

Here, $\mathbf{h}_v^{(t)}$ is the feature vector of node $v$ after $t$ layers, $W_1^{(t)}, W_2^{(t)} \in \mathbb{R}^{d_{t-1} \times d_t}$, $\mathbf{b}^{(t)} \in \mathbb{R}^{d_t}$ denotes the parameters of the $t$-th layer of the GNN, and $\sigma$ is some non-linear activation (e.g ReLU). It was shown in Morris et al. (2019) that GNNs composed from these layers have maximal expressive power with respect to all message-passing neural networks. In the experimental section we also experiment with Graph Isomorphism Network (GIN) (Xu et al., 2018). For further details on GNNs see Appendix A. In this work we use the most expressive GNN variants that use the "sum" aggregation function. Using a "max" or "mean" reduces the expressive power of the network, making it not powerful enough to solve simple counting problems (e.g. counting edges or computing node degrees). On the other hand, these networks give rise to slightly different definitions of patterns and can generalize better in some cases as shown in (Veličković et al., 2019), yet still suffer from size overfit. Exploring these networks is beyond the scope of this work.

### 3.1 SIZE GENERALIZATION IN SINGLE-LAYER GNNS

We start our discussion on size generalization with a theoretical analysis of a simple setup. We consider a single-layer GNN and an easy task and show that: (1) The training objective has many different solutions, but only a small subset of these solutions generalizes to larger graphs (2) Simple regularization techniques cannot mitigate the problem. This subsection serves as a warm-up for the next subsections that contain our main results.

Assume we train on a distribution of graphs with a fixed number of nodes $n$ and a fixed number of edges $m$. Our goal is to predict the number of edges in the graph using a 1-GNN with a single linear layer and additive readout function, for simplicity also consider the squared loss. The objective boils down to the following function for any graph $G$ in the training set:: $L(w_1, w_2, b; G) = \left( \sum_{u \in V(G)} \left( w_1 \cdot x_u + \sum_{v \in \mathcal{N}(u)} w_2 \cdot x_v + b \right) - y \right)^2$ . Here, $G$ is an input graph, $V(G)$ are the nodes of $G$, $\mathcal{N}(v)$ are all the neighbors of node $v$, $w_1, w_2$ and $b$ are the trainable parameters, $y$ is the target ($m$ in this case) and $x_v$ is the node feature for node $v$. Further, assume that we have no additional information on the nodes, so we can just embed each node as a

one-dimensional feature vector with a fixed value of 1. In this simple case, the trainable parameters are also one-dimensional. We note that the training objective can also be written in the following form $L(w_1, w_2, b; G) = (nw_1 + 2mw_2 + nb - m)^2$, and that one can easily find its solutions space, which is an affine subspace defined by $w_2 = \frac{m-n(w_1+b)}{2m}$. In particular, the solutions with $b + w_1 = 0$, $w_2 = 1/2$ are the only ones which do not depend on the specific training set graph size $n$, and generalize to graphs of any size. It can be readily seen that when training the model on graphs of fixed size (fixed $m, n$), gradient descent will have no reason to favor one solution over another and we will not be able to generalize. We also note that the generalizing solution is not always the least norm solution (with respect to both $L_1$ and $L_2$ norms) so simple regularization will not help here. On the other hand, it is easy to show that training on graphs with different number of edges will favor the generalizing solution. As we will see next, the problem gets worse when considering GNNs with multiple non-linear layers, and this simple solution will not help in this case: we can train deeper GNNs on a wide variety of sizes and the solution will not generalize to other sizes.

## 3.2 $d$-PATTERNS

We wish to understand theoretically when does a GNN which was trained on on graphs with a small number of nodes can generalize to graphs with a large number of nodes. To answer that question, we first analyze what information is received by each node in the graph from its neighboring nodes after a graph is processed by a GNN with $T$ layers. It is easy to see that every node can receive information about its neighbors which are at most $T$ hops away. We also know that nodes do not have full information about their order $T$ environment. For example, GNNs cannot determine if a triangle is present in a neighborhood of a given node Chen et al. (2020). In order to characterize the exact information that can be found in each node after a $T$ layer GNN, we use the definition of the WL test, specifically its iteration structure, which has the same representational power as GNNs (see Xu et al. (2018); Morris et al. (2019)), For more details on the WL test see Appendix A.

**Definition 3.1** ($d$-patterns). *Let $C$ be a finite set of node features and $N \in \mathbb{N}$. For $d \geq 0$ we define the set of $d$-**patterns** $P_d$ on graphs with maximal degree $N$ and node features from $C$. The definition is recursive in the following way: For $d = 0$, $P_0 = C$. We define $P_d$ to be the set of all tuples $(a, b)$ where $a \in P_{d-1}$ and $b$ is in multisets of size at most $N$ consisting of elements from $P_{d-1}$.*

*Let $G = (V, E)$ be a graph with maximal degree $N$ and a node feature $c_v \in C$ for every node $v \in V$. We define the $d$-pattern of a node $v \in V$ for $d \geq 0$ recursively: For $d = 0$, its 0-pattern is $c_v$. For $d > 0$ we say that $v$ with $\ell$ neighboring $d - 1$ patterns has a $d$-pattern $p = (p_v, \{(p_{i_1}, m_{p_{i_1}}), \ldots, (p_{i_\ell}, m_{p_{i_\ell}})\})$ iff node $v$ has $(d-1)$-pattern $p_v$ and for every $j \in \{1, \ldots, \ell\}$ the number of neighbors of $v$ with $(d-1)$-pattern $p_{i_j}$ is exactly $m_{p_{i_j}}$.*

The $d$-pattern of a node is an encoding of the $(d-1)$-patterns of itself and of its neighbors. For example, assume a graph has a maximal degree of $N$ and all the nodes start with the same node feature. The 1-pattern of each node is its degree. The 2-pattern of each node is for each possible degree $i \in \{1, \ldots, N\}$ the number of neighbors with degree $i$. In the same manner, the 3-pattern of a node is for each possible 2-pattern, the number of its neighbors with this exact 2-pattern. The definition of $d$-patterns can naturally be extended to the case of unbounded degrees. We have the following theorem which connects the $d$-patterns with the expressive power of GNNs:

**Theorem 3.2.** *Any function that can be represented by a d-layer GNN is constant on d-patterns.* In particular, the theorem shows that for any two graphs (of any size) and two nodes, one in each graph, if the nodes have the exact same $d$-pattern, then any $d$-layer GNN will output the same result for the two nodes. The full proof can be found in Appendix B, and follows directly from the analogy between the WL algorithm (see Appendix A) and $d$-patterns. Thm. 3.2 implies that $d$-patterns don't represent more expressive power than GNN. In the next subsection, we prove that GNNs can exactly compute $d$-patterns, and show that this capacity is tightly related to size generalization. It is also easy to see from the definition of $d$-patterns and the proof of Theorem 2 from Morris et al. (2019) that $d$-patterns exactly represent the expressive power of GNNs (with additive aggregation), thus this definition is a natural tool to study the properties of GNNs, such as size generalization.

## 3.3 GNNs MAY OVERFIT $d$-PATTERNS

We can now connect the size generalization problem to the concept of $d$-patterns. We start with an example: consider a node prediction task in which an output is specified for each node in an input

graph, and is solvable by a $d$-layer GNN. To perfectly solve this task, the model should produce the correct output for the $d$-pattern of all the nodes in the training set. Testing this GNN on a different set of graphs will succeed if the test set has graphs with similar $d$-patterns to those in the training set. Note that this requirement is not related to the *size* of the graphs but to the distribution of the $d$-patterns of the nodes in the test set.

In the following theorem we show rigorously, that given a set of $d$-patterns and output for each such pattern, there is an assignment of weights to a GNN with $O(d)$ layers that perfectly fits the output for each pattern. We will then use this theorem in order to show that, under certain assumptions on the distribution of $d$-patterns of the large graphs, GNNs can perfectly solve a task on a set of small graphs, and completely fail on a set on large graphs. In other words, we show that there are multiple global minima for the training objective that do not generalize to larger graphs.

**Theorem 3.3.** *Let $C$ be a finite set of node features, $P$ be a finite set of $d$-patterns on graphs with maximal degree $N \in \mathbb{N}$, and for each pattern $p \in P$ let $y_p \in [-1, 1]$ be some target label. Then there exists a 1-GNN $F$ with $d + 2$ layers, width bounded by $\max\left\{ (N+1)^d \cdot |C|, 2\sqrt{|P|} \right\}$ and ReLU activation such that for every graph $G$ with nodes $v_1, \ldots, v_n$, corresponding $d$-patterns $p_1, \ldots, p_n \subseteq P$ and node features from $C$, the output of $F$ on node $v_i$ is exactly $y_{p_i}$.*

The full proof is in Appendix B. Note that the width of the required GNN from the theorem is not very large if $d$ is small, where $d$ represents the depth of the 1-GNN. In practice, shallow GNNs are very commonly used and are proven empirically successful, while training deep GNNs was shown to be hard due to many problems like over-smoothing (Zhao and Akoglu, 2019).

Using the above theorem we can claim that there are assignments of weights to GNN that cannot "size-generalize", that is, given a specific task, the GNN succeeds on the task for small graphs (up to some bound) and fails on larger graphs, as long as there is a notable discrepancy between their $d$-patterns distributions:

**Corollary 3.4.** *Let $P_1$ and $P_2$ be distributions of small and large graphs respectively with finite support, and let $P_1^{d-pat}$ be the distribution of $d$-patterns over small graphs and similarly $P_2^{d-pat}$ for large graphs. For any node prediction task which is solvable by a 1-GNN with depth $d$ and $\epsilon > 0$ there exists a 1-GNN with depth at most $d + 2$ that has 0-1 loss smaller then $\epsilon$ on $P_1$ and 0-1 loss $\Delta$ on $P_2$, where*

$$\Delta(\epsilon) = \max_{A : P_1^{d-pat}(A) < \epsilon} P_2^{d-pat}(A). \tag{1}$$

*Here, $A$ is a set of $d$-patterns and $P(A)$ is the total probability mass for that set under $P$.*

Intuitively, large $\Delta$ means that there exists a set of d-patterns that have a low probability for small graphs and high probability for large graphs. Corollary 3.4 implies that the major factor in the success of GNN to generalize to larger graphs is not the graph size, but the distribution of the $d$-patterns. Different distributions of $d$-patterns lead to large $\Delta$ and thus to bad generalization to larger graphs. On the other hand, from Thm. 3.2 we immediately get that similar distributions of $d$-patterns imply that *every* GNN model that succeeds on small graphs will also succeed on large graphs, since GNNs are constant on $d$-patterns:

**Corollary 3.5.** *In the setting of Corollary 3.4, also assume that all the patterns that have a positive probability in $P_2^{d-pat}$ also have a positive probability in $P_1^{d-pat}$. Then, for any node prediction task solvable by a depth $d$ GNN, any $1 - GNN$ that have $0$ loss (w.r.t the $0 - 1$ loss) on $P_1$ will also have $0$ loss on $P_2$.*

**Examples.** Corollary 3.4 shows that even for simple tasks, GNN may fail, here are two simple examples. (i) Consider the task of calculating the node degree. From Corollary 3.4 there is a GNN that successfully output the degree of nodes with max degree up to $N$ and fails on nodes with larger degrees. Note that this problem can easily be solved for any node degree with a 1-layer GNN. (ii) Consider some node regression task, when the training set consists of graphs sampled i.i.d from an Erdos-Renyi graph $G(n, p)$ [1], and the test set contains graphs sampled i.i.d from $G(2n, p)$. In this case, a GNN trained on the training set will be trained on graphs with an average degree $np$, while the test set contains graphs with an average degree $2np$. This means that the $d$-patterns in the train and test set are very different, and by Corollary 3.4 the GNN may overfit.

---

[1]Graphs with $n$ nodes such that each edge exists with probability $p$.

**Graph prediction tasks.** Our theoretical results discuss node prediction tasks. We note that they are also relevant for graph prediction tasks where there is a single output to each input graph. The reason for that is that in order to solve graph prediction tasks, a GNN first calculates node features and then pools them into a single global graph feature. Our analysis shows that the first part of the GNN, which is responsible for calculating the node features, might not generalize to large graphs. As a result, the GNN will generate an uninformative global graph feature and the GNN will fail on the original graph prediction task. In the experimental sections, we show that the size generalization problem is indeed relevant for both node and graph prediction tasks. Here is a formal statement regarding graph prediction tasks, the full proof can be found in Appendix B.

**Corollary 3.6.** *Let $P_1$ and $P_2$ be distributions of small and large graphs respectively with finite support. Let $P_1^{d-pat}$ be the distribution of $d$-patterns over small graphs and similarly $P_2^{d-pat}$ for large graphs, and assume that the supports of $P_1^{d-pat}$ and $P_2^{d-pat}$ are disjoint. For any graph prediction task solvable by a 1-GNN with depth $d$ and summation readout function, there exists a 1-GNN with depth at most $d + 3$ that perfectly solves the task on $P_1$ and fails on all graphs from $P_2$.*

**Relation to Morris et al. (2019); Xu et al. (2018).** We note that Theorem 3.3 and Corollary 3.4 are somewhat related to the expressivity results in (Xu et al., 2018; Morris et al., 2019) that show that GNNs can be as powerful as the WL test. Here, we show that the expressive power of GNNs can cause negative effects when there is a discrepancy between the training and test sets.

### 3.4 EMPIRICAL VALIDATION

In the previous subsection we have shown that for any node task, and any two datasets of graphs with different sizes that significantly differ in their $d$-patterns distributions, there is a 1-GNN that successfully solves the task on one dataset but fails on the second. In this subsection, we show empirically that reaching these "overfitting" GNNs is actually very common. Specifically, the size-overfit phenomenon is prevalent when the $d$-patterns of in the large graph distribution are not found in the small graph distribution. We also show that GNNs can generalize to larger graphs if the distribution of $d$-patterns remains similar to the distribution of patterns in the small graphs.

To show this, we use a controlled regression task in a student-teacher setting. In this setting, we sample a "teacher" GNN with random weights, freeze the network, and label each graph in the dataset using the output of the "teacher" network. Our goal is to train a "student" network, which has the same architecture as the "teacher" network, to fit the labels of the teacher network. The advantages of this setting are two-fold: (1) *A solution is guaranteed to exist*: We know that there is a weight assignment of the student network which perfectly solve the task for graphs of any size. (2) *Generality*: It includes all tasks solvable by constant depth GNNs. We discuss more settings below.

**Architecture and training protocol.** We use 1-GNN as defined in (Morris et al., 2019). The number of GNN layers in the network we use is either $1, 2$ or $3$; the width of the teacher network is 32, and of the student network 64, providing more expressive power to the student network. We obtained similar results when testing with a width of 32, same as the teacher network. We use a summation readout function followed by a two-layer fully connected suffix. We use ADAM with learning rate $10^{-3}$. We performed a hyper-parameters search on the learning rate and weight decay and use validation-based early stopping on the source domain (small graphs). The results are averaged over 10 random seeds. All runs used Pytorch Geometric (Fey and Lenssen, 2019) on NVIDIA DGX-1.

**Results.** Fig. 1 compares the loss of GNNs as the distribution of $d$-patterns changes, for the task of teacher-student graph level regression. The model was trained on graphs generated using the $G(n, p)$ model. We show the normalized $L_2$ loss computed on test, where output is normalized by the average test-set (target) output. The left panel shows the test loss when training on $n \in [40, 50]$ and $p = 0.3$ and testing on $G(n, p)$ graphs with $n = 100$ and $p$ varying from 0.05 to 0.5. In this experiment, the expected node degree is $np$, hence the distribution of $d$-patterns is most similar to the one observed in the training set when $p = 0.15$. Indeed, this is the value of $p$ where the test loss is minimized. The right panel is discussed in the caption. These results are consistent with Corollary 3.4, since when the distributions of $d$-patterns are far the model is not able to generalize well, and it does generalize well when these distributions are similar.

To give further confirmation to the effect of the local distributions on the generalization capabilities of GNN we conducted the following two experiments: (1) We tested on the teacher-student setup with a 3-layer GNN on graphs of sizes uniformly from $n = 40, \ldots, 50$ and sampled from $G(n, 0.3)$.

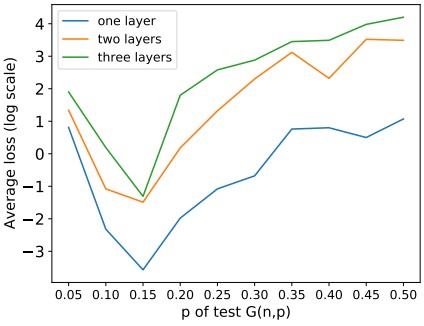 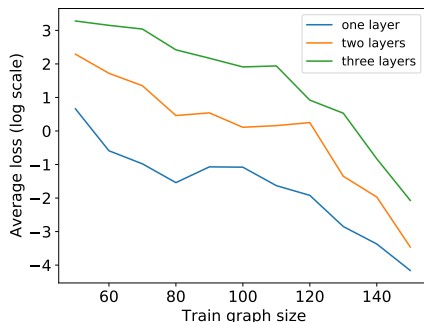

Figure 1: The effect of graph size and $d$-pattern distribution on generalization in G(n,p) graphs. **(left) The effect of distribution of $d$-patterns**. Train on $n$ drawn uniformly from $[40, 50]$ and $p = 0.3$ test on $n = 100$ and varying $p$ ; **(right) The effect of train-graph size.** Train on $n$ drawn uniformly from $[40, x]$ where $x$ varies and $p = 0.3$; test on $n = 150$, $p = 0.3$.

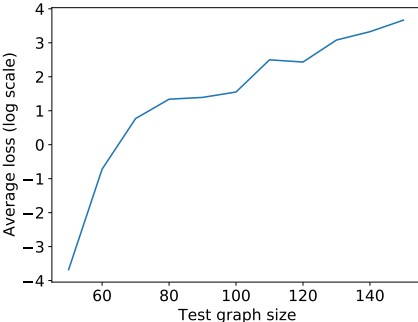 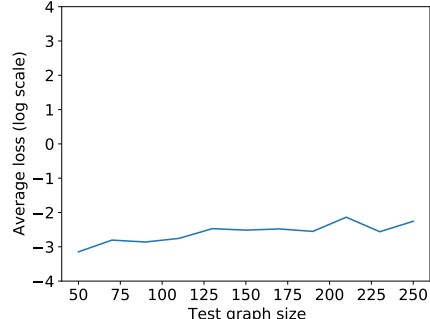

Figure 2: Constant training size and varying test size with and without normalization. **Left**: constant $p$ which leads to different $d$-patterns. **Right**: $p$ is normalized to keep $d$-patterns distribution similar.

We tested on graphs sampled from $G(N, 0.3)$ where $N$ varies from 50 up to 150. It is evident that as the graph size in the test set increases, the model performs worse. (2) We did the same test as in (1), but this time we normalize $p$ on the test set so that $p \cdot N = 15$, which is the approximate ratio for the training set. Here we even went further to train up to sizes $N = 250$. In this experiment, the GNN successfully generalized to larger graphs, since the local distributions of the train and test set are indeed very similar. For the results see Fig. 2.

We also tested on the tasks of finding the max clique in a graph, calculating the number of edges, and the node prediction tasks of calculating the node degree, and the student-teacher task at the node level. In addition, we tested on the popular GIN architecture (Xu et al., 2018), and show that the size generalization problem also occurs there. We also tested on ReLU, Tanh, and sigmoid activations. See additional experiments in Appendix C.

## 4 TOWARDS IMPROVING SIZE-GENERALIZATION

The results from the previous section show that the problem of size generalization is not only related to the size of the graph in terms of the number of nodes or edges but to the distribution of $d$-patterns induced by the distributions from which the graphs are sampled. Based on this observation, we now formulate the size generalization problem as a domain adaptation (DA) problem. We then build on techniques from domain adaptation and suggest two approaches to improve size generalization. (1) Self-supervised learning on the target domain (large graphs) and (2) Semi-supervised learning with a few labeled target samples. We consider the DA setting where we are given two distributions over graphs: a source distribution $\mathcal{D}_S$ (say, for small graphs) and a target distribution $\mathcal{D}_T$ (say, for large graphs). We consider two settings. First, the unlabeled DA setting, where we have access to labeled samples from the source $\mathcal{D}_S$ but the target data from $\mathcal{D}_T$ is unlabeled. Our goal is to infer labels on

a test dataset sampled from the target $\mathcal{D}_T$. Second, we consider a semi-supervised setup, where we also have access to a small number of labeled examples from the target $\mathcal{D}_T$.

**Size generalization with Self-supervised learning.** In Self-supervised learning (SSL) for DA, a model is trained on unlabeled data to learn a *pretext* task, which is different from the main task at hand. If the pretext task is chosen wisely, the model learns useful representations (Doersch et al., 2015; Gidaris et al., 2018) that can help with the main task. Here, we train the pretext task on both the source and target domains, as was done for images and point clouds (Sun et al., 2019; Achituve et al., 2020). The idea is that the pretext task aligns the representations of the source and target domains leading to better predictions of the main task for target graphs.

{For a detailed review of the training procedures and the losses see Appendix E. Given a pretext task we consider two different training procedures: (1) **Multi-task learning (MTL)**: parallel training of the main task on a source domain and a pretext task on both the source of target domain (You et al., 2020). In this case, the architecture consists of a main GNN that acts as a feature extractor and two secondary networks (heads) that operate on the extracted features and try to predict the main task and the pretext task. (2) **Pretraining (PT)**: in this procedure (Hu et al., 2019), the GNN feature extractor is trained until convergence on the pretext task on both the source and target examples. Then, the GNN part is frozen, and only the head of the model is trained on labeled examples from the source.

Figure 3: Left: a graph with node features represented by colors. Right: A tree that represents the $d$-patterns for the black node. The tree descriptor is the number of nodes from each class in each layer of the tree.

**Pattern-tree pretext task.** We propose a novel pretext task which is motivated by the definition of $d$-patterns. We do that by constructing a tree that fully represents the $d$-patterns of each node (see e.g., Xu et al. (2018)). We then calculate a descriptor of the tree, which is a vector containing counts of the number of nodes from each class in each layer of the tree. We treat this descriptor as the target label to be reconstructed by the SSL task. For more details see Figure 3. Intuitively, in order to be successful on a task on graphs, the GNN needs to correctly represent the pattern trees of the nodes of the graphs. This means, that to generalize to the target domain $\mathcal{D}_t$, the GNN needs to be forced to represent pattern trees from both the source and the target distributions. For more details about the construction of the pattern tree see Appendix D. In short, each tree corresponds to a $d$-pattern in the following way: the $d$-pattern tree of a node can be seen as a multiset of the children of the root, and each child is a multiset of its children, etc. The pattern tree is a different description of the $d$-pattern of a node. This means that a GNN that successfully represent a pattern tree also represents its corresponding $d$-pattern, thus connecting this SSL task to the theory from Sec. 3.

**Semi-supervised setup.** We also consider a case where a small number of labeled samples are available for the target domain. A natural approach is to train an SSL pretext task on samples from both the source and target domain, and train the main task on all the labeled samples available. We tested this setup with 1, 5, or 10 labeled examples from the target domain.

### 4.1 EXPERIMENTS

**Architecture and training protocol.** The setup is the same as in Subsection 3.4 with the following changes. We use a three-layer GNN in all experiments. Multi-task learning is used with equal weight to the main and SSL tasks. In the semi-supervised setup, we used an equal weight for the main task and the labeled examples from the target domain.

**Baselines.** We compare our new pretext task to the following baselines: (1) **Vanilla**: standard training on the source domain; (2) **HomoGNN** (Tang et al., 2020) a homogeneous GNN without the bias term trained on the source domain; (3) **Graph autoencoder (GAE)** pretext task (Kipf and Welling, 2016); (4) **Node masking (NM)** pretext task from Hu et al. (2019) where at each training iteration we mask $10\%$ of the node features and the goal is to reconstruct them. In case the graph does not have node features then the task was to predict the degree of the masked nodes. (5) **Node metric learning (NML)**: we use metric learning to learn useful node representations. We use a corruption function that given a graph and corruption parameter $p \in [0, 1]$, replaces $p|E|$ of the edges with random edges, and thus can generate positive ($p = 0.1$) and negative ($p = 0.3$) examples for all nodes of the graph. we train with the triplet loss (Weinberger and Saul, 2009).

| DATASETS | DEEZER | IMDB - B | NCI1 | NCI109 | PROTEINS | TWITCH | DD | AVERAGE |
|---|---|---|---|---|---|---|---|---|
| SMALL GRAPHS | $56.5 \pm 0.8$ | $63.2 \pm 3.3$ | $75.5 \pm 1.6$ | $78.4 \pm 1.4$ | $75.4 \pm 3.1$ | $69.7 \pm 0.2$ | $71.1 \pm 4.4$ | 70.0% |
| VANILLA | $41.1 \pm 6.8$ | $55.9 \pm 7.8$ | $65.9 \pm 4.3$ | $68.9 \pm 3.8$ | $76.0 \pm 8.5$ | $60.5 \pm 3.6$ | $76.3 \pm 3.2$ | 63.5% |
| HOMO-GNN | $40.5 \pm 6.6$ | $56.3 \pm 7.0$ | $66.0 \pm 3.7$ | $68.8 \pm 3.2$ | $77.1 \pm 10$ | $60.8 \pm 2.3$ | $\mathbf{76.8 \pm 3}$ | 63.8% |
| NM MTL | $\mathbf{51.6 \pm 8.5}$ | $55.6 \pm 6.8$ | $49.9 \pm 7.8$ | $61.7 \pm 5.7$ | $78.8 \pm 8.4$ | $49.5 \pm 2.8$ | $67.4 \pm 5.4$ | 59.2% |
| NM PT | $50.1 \pm 7.5$ | $54.9 \pm 6.7$ | $51.7 \pm 6.6$ | $55.8 \pm 5.0$ | $78.2 \pm 8.2$ | $48.4 \pm 4.0$ | $60.3 \pm 15.9$ | 57.1% |
| GAE MTL | $49.4 \pm 11.0$ | $55.5 \pm 6.0$ | $51.2 \pm 9.9$ | $57.6 \pm 9.4$ | $79.5 \pm 11.7$ | $62.5 \pm 5.1$ | $67.8 \pm 10.0$ | 60.5% |
| GAE PT | $47.1 \pm 10.0$ | $54.1 \pm 6.8$ | $58.9 \pm 7.6$ | $67.2 \pm 5.6$ | $70.5 \pm 9.4$ | $53.6 \pm 4.7$ | $69 \pm 7.1$ | 60.1% |
| NML MTL | $46.4 \pm 9.5$ | $54.4 \pm 7.0$ | $52.3 \pm 6.3$ | $56.2 \pm 6.5$ | $78.7 \pm 6.8$ | $57.4 \pm 4.1$ | $64.7 \pm 11.9$ | 58.6% |
| NML PT | $48.4 \pm 10.7$ | $53.8 \pm 6.1$ | $54.6 \pm 6.2$ | $56.1 \pm 8.1$ | $76.3 \pm 8.0$ | $54.9 \pm 4.7$ | $61.4 \pm 15.1$ | 57.9% |
| PATTERN MTL | $45.6 \pm 8.8$ | $56.8 \pm 9.2$ | $60.5 \pm 7.5$ | $67.9 \pm 7.2$ | $75.8 \pm 11.1$ | $61.6 \pm 3.5$ | $\mathbf{76.8 \pm 3}$ | 63.6% |
| PATTERN PT | $44 \pm 7.7$ | $\mathbf{61.9 \pm 3.2}$ | $\mathbf{67.8 \pm 11.7}$ | $\mathbf{74.8 \pm 5.7}$ | $\mathbf{84.7 \pm 5.1}$ | $\mathbf{64.5 \pm 3.3}$ | $74.9 \pm 5.2$ | 67.5% |

Table 1: Test accuracy of compared methods in binary classification tasks. The Pattern task with pretraining achieves the highest accuracy in most tasks and has 4% higher average accuracy than the second-best method. High variance is due to the domain shift between the source and target domain.

**Datasets.** We use datasets from Morris et al. (2020) and Rozemberczki et al. (2020) (Twitch egos and Deezer egos). We selected datasets that have a sufficient number of graphs (more than 1,000) and with a non-trivial split to small and large graphs as detailed in Appendix F.1. In total we used 7 datasets, 4 in molecular biology (NCI1, NCI109, D&D, Proteins), and 3 of social networks (Twitch ego nets, Deezer ego nets, IMDB-Binary). In all datasets, 50% smallest graphs were assigned to the training set, and the largest 10% of graphs assigned to the test set. We further split a random 10% of the small graphs as a validation set.

**Results.** Table 1 compares the effect of using the Pattern-tree pretext task to the baselines described above. The *small graphs* row presents vanilla results on a validation set with small graphs. The small graph accuracy on 5 out of 7 datasets is larger by 7.3%-15.5% than on large graphs, indicating that the size-generalization problem is indeed prevalent in real datasets. Pretraining with the $d$-patterns pretext task outperforms other baselines in 5 out 7 datasets, with an average 4% improved accuracy on all datasets. HOMO-GNN slightly improves over the vanilla while other pretext tasks do not improve average accuracy. Naturally, the accuracy here is much lower than SOTA on these datasets because the domain shift makes the problem much harder. In Appendix F.2 we show the 1-pattern distribution discrepancy between large and small graphs in two real datasets: IMDB (large discrepancy) and D&D (small discrepancy). Correspondingly, the pattern tree SSL task improved performance on the IMDB dataset, while not improving performance on the D& D dataset. This gives further evidence that a discrepancy between the $d$-patterns leads to bad generalization, and that correctly representing the patterns of the test set can improve performance.

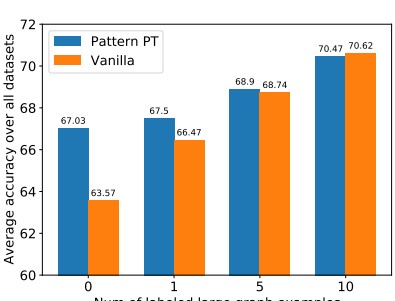

Figure 4: Mean accuracy over all datasets in Tab. 1 for $d$-pattern pre training and no SSL (Vanilla).

Figure 4 compares the performance of vanilla training versus pretraining with the pattern-tree pretext task in the semi-supervised setup. The accuracy monotonically increases with respect to the number of labeled examples in both cases. Moreover, pretraining with the pretext task yields better results in the case of 0,1,5 labeled examples and comparable results in the case we use 10 labeled examples. We additionally tested on the synthetic tasks discussed in Sec. 3, and show that the pattern-tree pretext task improves in the student-teacher setting, while it does not solve the edge count or degree prediction tasks. On the other hand, adding even a single labeled sample from the target distribution significantly improves performance on the synthetic tasks we tested on. For more details see Sec. F.

## 5   CONCLUSION AND DISCUSSION

This paper is a first step towards understanding the size generalization problem in graph neural networks. We showed that GNNs do not naturally generalize to larger graphs even on simple tasks, characterized how this failure depends on $d$-patterns, and suggested two approaches that can improve generalization. Our characterization of $d$-patterns is likely to have implications to other problems where generalization is harmed by distribution shifts, and offer a way to mitigate those problems.

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

## A  PRELIMINARIES

This section discusses two key concepts that we use throughout the paper: (1) Graph Neural Networks; (2) The Weisfeiler-Lehman graph isomorphism test. At the end of the section, we also discuss the setup of constant GNN depth that we use in this paper.

**Notation.** We denote by $\{(a_1, m_{a_1}), \ldots, (a_n, m_{a_n})\}$ a **multiset**, that is a set where we allow multiple instances of the same element. Here $a_1, \ldots, a_n$ are distinct elements, and $m_{a_i}$ is the number of times $a_i$ appears in the multiset. bold-face letters represent vectors.

### A.1  GRAPH NEURAL NETWORKS

We consider the common message-passing GNN architecture (38) defined as follows: Let $G = (V, E)$ be a graph, and for each node $v \in V$ let $\mathbf{h}_v^{(0)} \in \mathbb{R}^{d_0}$ be a node feature vector. Then for every $t > 0$ we define:

$$\mathbf{h}_v^{(t)} = \text{UPDATE}\left(\mathbf{h}_v^{(t-1)}, \text{AGG}\left(\left\{\mathbf{h}_u^{(t-1)} : u \in \mathcal{N}(v)\right\}\right)\right).$$

Here, AGG is an permutation invariant aggregation function such as summation, averaging or taking max, UPDATE is an update function such as an MLP, and $\mathcal{N}(v)$ denotes the set of neighbors of node $v$. In case the graph is directed then $\mathcal{N}(v)$ denotes the set of nodes with edges incoming to $v$. For node prediction tasks, the output of a $T$-layer GNN for node $v$ is $\mathbf{h}_v^{(T)}$, while for graph prediction tasks an additional readout layer is used:

$$g^{(T)} = \text{READOUT}\left\{\mathbf{h}_v^{(T)} : v \in V\right\},$$

where READOUT is some invariant function such as summation, averaging or taking the max, possibly followed by a fully connected network.

In our theoretical results, we focus on the **first order GNN** architecture from (25) defined in the following way:

$$\mathbf{h}_v^{(t)} = \sigma\left(W_2^{(t)}\mathbf{h}_v^{(t-1)} + \sum_{u \in \mathcal{N}(v)} W_1^{(t)}\mathbf{h}_u^{(t-1)} + \mathbf{b}^{(t)}\right).$$

Here, $W_1^{(t)}, W_2^{(t)} \in \mathbb{R}^{d_{t-1} \times d_t}$, $\mathbf{b}^{(t)} \in \mathbb{R}^{d_t}$ denotes the parameters of the $t$-th layer of the GNN, and $\sigma$ is some non-linear activation (e.g ReLU). It was shown in (25) that GNNs composed from these layers have maximal expressive power with respect to all message passing neural networks. In general, we focus on GNNs that use the popular sum aggregation as it has maximal expressive power with respect to other aggregations used in message passing GNNs (38). We discuss other aggregation functions and the trade-of they present in Section 5.

### A.2  THE WEISFEILER-LEHMAN TEST

The definition of $d$-patterns, which are used frequently in this paper, is closely related to the Weisfeiler-Lehman (WL) test (37; 13). The WL test is an algorithm to test whether two graphs are isomorphic and was recently coupled with the expressive power of GNNs. (38; 25). It is easiest to describe WL by using an equivalent algorithm called the color-refinement algorithm (13). The color refinement algorithm is executed sequentially, wherein each stage the algorithm generates a

descriptor for each node according to the descriptors of its direct neighbors. The features in each stage can be used to define equivalence classes of nodes and the process continues until these equivalence classes cannot be refined anymore. The final graph descriptor is a histogram of the node features, and is closely related to the definition of $d$-patterns.

## A.3 CONSTANT DEPTH VS. ADAPTIVE DEPTH TASKS

As stated earlier, multiple factors control size generalization. One important factor is the type of task: We make a distinction between two types of graph tasks. (1) Tasks that can be solved by constant depth GNN; A good example for such task is determining whether a certain, constant size connectivity pattern is found in the graph (2) Tasks that require the depth of the GNN to be related to different parameters of the problem, such as the diameter of the graph. To exemplify, the task of calculating the diameter of a graph can be solved with a GNN only if its depth depends on the size of the graph (For more details see (32; 21)). On the other hand, tasks that require local information about each node can be solved with a constant depth GNN. We note that although there are several recent works about adaptive depth GNN, e.g. (32; 27), most currently used GNNs have constant and relatively small depth (7)). In this paper, we focus on the first kind of tasks that require a constant number of GNN layers. We note that the size-generalization problem we discuss in this paper is also relevant for the second type of tasks, but we chose to discuss the problem in the simpler setup for clarity.

## B PROOFS FROM SEC. 3

*Proof of Thm. 3.2.* We show that from the definition of $d$-patterns, and the 1-WL algorithm (see (37)), the color given by the WL algorithm to two nodes is equal iff their $d$-pattern is equal. For the case of $d = 0$ it is clear. Suppose it is true for $d - 1$, the WL algorithm at iteration $d$ give node $v$ a new color based on the colors given in iteration $d - 1$ to the neighbors of $v$. This means, that the color of $v$ at iteration $d$ depends on the multiset of colors at iteration $d - 1$ of its neighbors, which by induction is the $d - 1$-pattern of the neighbors of $v$. To conclude, we use Theorem 1 from (25) which shows that GNNs are constant on the colors of WL, hence also constant on the $d$-patterns. □

To prove Thm. 3.3 we will first need to following claim from (40) about the memorization power of ReLU networks:

**Theorem B.1.** *Let $\{\mathbf{x}_i, y_i\}_{i=1}^N \in \mathbb{R}^d \times \mathbb{R}$ such that all the $\mathbf{x}_i$ are distinct and $y_i \in [-1, 1]$ for all $i$. Then there exists a 3-layer fully connected ReLU neural network $f : \mathbb{R}^d \to \mathbb{R}$ with width $2\sqrt{N}$ such that $f(\mathbf{x}_i) = y_i$ for every $i$.*

We will also need the following lemma which will be used in the construction of each layer of the 1-GNN:

**Lemma B.2.** *Let $N \in \mathbb{N}$ and $f : \mathbb{N} \to \mathbb{R}$ be a function defined by $f(n) = \mathbf{w}_2^\top \sigma(\mathbf{w}_1 n - \mathbf{b})$ where $\mathbf{w}_1, \mathbf{w}_2, \mathbf{b} \in \mathbb{R}^N$, and $\sigma$ is the ReLU function. Then for every $y_1, \dots, y_N \in \mathbb{R}$ there exists $\mathbf{w}_1, \mathbf{w}_2, \mathbf{b}$ such that $f(n) = y_n$ for $n \leq N$ and $f(n) = (n - N + 1)y_N - (n - N)y_{N-1}$ for $n > N$.*

*Proof.* Define $\mathbf{w}_1 = \begin{pmatrix} 1 \\ \vdots \\ 1 \end{pmatrix}$, $\mathbf{b} = \begin{pmatrix} 0 \\ 1 \\ \vdots \\ N - 1 \end{pmatrix}$. Let $a_i$ be the $i$-th coordinate of $\mathbf{w}_2$, we will define $a_i$ recursively in the following way: Let $a_1 = y_1$, suppose we defined $a_1, \dots, a_{i-1}$, then define $a_i = y_i - 2a_{i-1} - \cdots - ia_1$. Now we have for every $n \leq N$:

$$f(n) = \mathbf{w}_2^\top \sigma(\mathbf{w}_1 n - \mathbf{b}) = na_1 + (n - 1)a_2 + \cdots + a_n = y_n \ .$$

For $n > N$ we can write $n = N + k$ for $k \geq 1$, then we have:

$$\begin{aligned}
f(n) = \mathbf{w}_2^\top \sigma(\mathbf{w}_1(N + k) - \mathbf{b}) &= (N + k)a_1 + (N + k - 1)a_2 + \cdots + (k + 1)a_N \\
&= y_N + k(a_1 + a_2 + \cdots + a_N) = y_N + k(y_N - a_{N-1} - 2a_{N-2} - \cdots - (N - 1)a_1) \\
&= (k + 1)y_N - ky_{N-1}
\end{aligned}$$

$\square$

Now we are ready to prove the main theorem:

*Proof of Thm. 3.3.* We assume w.l.o.g that at the first iteration each node $i$ is represented as a one-hot vector $\mathbf{h}_i^{(0)}$ of dimension $|C|$, with its corresponding node feature. Otherwise, since there are $|C|$ node features we can use one GNN layer that ignores all neighbors and only represent each node as a one-hot vector. We construct the first $d$ layers of the 1-GNN $F$ by induction on $d$. Denote by $a_i = |C| \cdot \left(N^i + N^{i-1}\right)$ the dimension of the $i$-th layer of the GNN for $1 \leq i \leq d$, and $a_0 = |C|$.

The mapping has two parts, one takes the neighbors information and maps it into a feature representing the multiset of $d$-patterns, the other part is simply the identity saving information regarding the d-pattern of the node itself.

The $d$ layer structure is

$$\mathbf{h}_v^{(d)} = U^{(d+1)}\sigma\left(W_2^{(d)}\mathbf{h}_v^{(d-1)} + \sum_{u \in \mathcal{N}(v)} W_1^{(d)}\mathbf{h}_u^{(d-1)} - \mathbf{b}^{(d)}\right)$$

We set $W_2^{(d)} = [0, I]^T$, $W_1^{(d)} = [\tilde{W}_1^{(d)T}, 0]^T$ and $U^{(d+1)} = [\tilde{U}^{(d+1)T}, 0]^T$ with $\tilde{W}_1^{(d)} \in \mathbb{R}^{Na_{d-1} \times a_{d-1}}$ and $\tilde{U}^{(d+1)} \in \mathbb{R}^{Na_{d-1} \times Na_{d-1}}$. For $\tilde{W}_1^{(d)}$ we set $\mathbf{w}_i^{(1)}$, its $i$-th row, to be equal to $\mathbf{e}_n$ where $n = \left\lceil \frac{i}{N} \right\rceil$. Let $b_i^{(1)}$ be the $i$-th coordinate of $\mathbf{b}^{(d)}$, be equal to $i - 1 \pmod{N}$ for $i \leq N \cdot a_{d-1}$ and zero otherwise. What this does for the first $a_{d-1}$ elements of $W_2^{(d)}\mathbf{h}_v^{(d)} + \sum_{u \in \mathcal{N}(v)} W_1^{(d)}\mathbf{h}_u^{(d)}$ is that each dimension $i$ hold the number of neighbors with a specific (d-1)-pattern. We then replicate this vector N times, and for each replica we subtract a different bias integer ranging from 0 to $N - 1$. To that output we concatenate the original $\mathbf{h}_v^{(d-1)}$

Next we construct $\tilde{U}^{(d+1)} \in \mathbb{R}^{Na_{d-1} \times Na_{d-1}}$ in the following way: Let $\mathbf{u}_i^{(d+1)}$ be its $i$-th row, and $u_{i,j}^{(d+1)}$ its $j$-th coordinate. We set $u_{i,j}^{(d+1)} = 0$ for every $j$ with $j \neq \left\lceil \frac{i}{N} \right\rceil$ and the rest $N$ coordinates to be equal to the vector $\mathbf{w}_2$ from Lemma B.2 with labels $y_\ell = 0$ for $\ell \in \{1, \ldots, N\} \setminus \{(i \bmod N) + 1\}$ and $y_\ell = 1$ for $\ell = (i \bmod N) + 1$.

Using the above construction we encoded the output on node $v$ of the first layer of $F$ as a vector:

This encoding is such that the $i$-th coordinate of $\mathbf{h}_v^{(d+1)}$ for $1 \leq i \leq N \cdot a_{d-1}$ is equal to 1 iff node $v$ have $(i \bmod N) + 1$ neighbors with node feature $\left\lceil \frac{i}{N} \right\rceil \in \{1, \ldots, |C|\}$. The last $a_{d-1}$ rows are a copy of $\mathbf{h}_v^{(d-1)}$.

**Construction of the suffix.** Next, we construct the last two layers. First we note that for a node $v$ with $d$-pattern $p$ there is a unique vector $\mathbf{z}_p$ such that the output of the 1-GNN on node $v$, $\mathbf{h}_v^{(d)}$, is equal to $\mathbf{z}_p$. From our previous construction one can reconstruct exactly the (d-1)-pattern of each node, and the exact number of neighbors with each (d-1)-pattern and therefore can recover the d-pattern correctly from the $\mathbf{h}_v^{(d)}$ embedding.

Finally, we use Thm. B.1 to construct the 3-layer fully connected neural network with width at most $2\sqrt{|P|}$ such that for every pattern $p_i \in P$ with corresponding unique representation $\mathbf{z}_{p_i}$ and label $y_i$, the output of the network on $\mathbf{z}_{p_i}$ is equal to $y_i$. We construct the last two layers of the 1-GNN such that $W_1^{(d+1)}, W_1^{(d)} = 0$, and $W_2^{(d+1)}, \mathbf{b}^{(d+1)}, W_2^{(d+2)}, \mathbf{b}^{(d+2)}, W^{(d+3)}$ are the matrices produced from Thm. B.1. Note that $W^{(d+3)}$ is the linear output layer constructed from the theorem, thus the

final output of the 1-GNN for node $v$ is $W^{(d+3)} \cdot h_v^{(d+2)}$, where $h_v^{(d+2)}$ is the output after $d+2$ layers. ☐

*Proof of Corollary 3.4.* By the assumption, the output of the task is determined by the $d$-patterns of the nodes. For each node with pattern $p_i$ let $y_i$ be the output of the node prediction task. Define

$$A = \arg \max_{A': P_1^{d-pat}(A') < \epsilon} P_2^{d-pat}(A') \tag{2}$$

By Thm. 3.3 there exists a 1-GNN such that for any $d$-pattern $p_i \in A$ gives a wrong label and for every pattern outside $A$, $p_j \in A^c$ gives the correct label. Note that we can use Thm. 3.3 since both $A$ and $A^c$ are finite, because we assumed that distributions on the graphs have finite supports. The 0-1 loss for small and large graphs is exactly $P_1^{d-pat}(A)$ and $P_2^{d-pat}(A)$ respectively.

☐

*Proof of Corollary 3.6.* Let $D_1$ and $D_2$ be all the $d$-patterns that appear in graphs from $P_1$ and $P_2$ respectively. By the assumption that the task is solvable by a $d$-layer GNN, and the same reasoning as in the proof of Corollary 3.4, the output of the GNN which solves the task depends only on the $d$-patterns. Thus, by Thm. 3.3 there exists a $d+2$-layer GNN that perfectly solves the task on graphs with patterns from $D_1$ and in addition sends every pattern from $D_2$ to zero. If zero is not a solution of the task for any graph from $P_2$ we are finished. Otherwise, if zero is a solution of a graph from $P_2$ but not for a graph from $P_1$ we can add an MLP after the summation readout function that outputs the identity on every input except zero, for which it outputs a value which is not a solution for the task for any graph in $P_2$. Note that this is possible since $P_2$ has finite support. In the case that zero is also a solution of the task on a graph from $P_1$, let $n_1, \ldots, n_\ell$ be the sizes of graphs from $P_2$. We pick some $y \in \mathbb{R}$ such that $y \cdot n_i$ is not an output of any graph from $P_1$ and $P_2$, there is such $y$ since the distributions have a finite support. Now, we can change the proof so that the output of the GNN on all patterns from $D_2$ is equal to $y$. Note that now the output of any graph from $P_2$ after the summation readout function is not the correct output. In case there is an MLP after the readout function, the proof can be readily changed to account for that. ☐

*Proof or Corollary 3.5.* Let $h$ be a 1-GNN that have $0$ loss on $P_1$. In particular, for any pattern in $P_1^{d-pat}$, $h$ successfully predict the correct label. By the assumption, the patterns that appear in $P_2^{d-pat}$ are contained in the pattern that appear in $P_1^{d-pat}$. Using Thm. 3.2, that any 1-GNN is constant on $d$-patterns, we get that $h$ will also succeed on the pattern from $P_2^{d-pat}$, hence will have $0$ loss on $P_2$. ☐

## C  ADDITIONAL EXPERIMENTS FROM SEC. 3.4

First, we consider the max-clique problem. The goal of this problem is given a graph, to output the size of the maximal clique. This is in general an NP-hard problem, hence a constant depth GNN will not be able to solve it for graphs of all sizes. For this task we sampled both the train and test graphs from a geometrical distribution defined as follows: given a number of nodes $n$ and radius $\rho$ we draw $n$ points uniformly in $[0, 1]^2$, each point correspond to a node in the graph, and two nodes are connected if their corresponding points have a distance less than $\rho$. We further analyzed the effects of how the network depth and architecture affect size generalization.

Table 2 presents the test loss on the max-clique problem. Deeper networks are substantially more stricken due to the domain shift. If the test domain has a similar pattern distribution, increasing the neural network depth from one layer to three layers results in a small decrement of at most $20\%$ to the loss. However, if the pattern distribution is different than the train pattern distribution, such change may increase the loss by more than $2.5\times$. We also show that the problem is consistent in both 1-GNN and GIN architectures.

| $\rho_{train}/\rho_{test}$ | 1-GNN | | | GIN | | |
|---|---|---|---|---|---|---|
| | 1-layer | 2-layers | 3-layers | 1-layer | 2-layers | 3-layers |
| 1 | $22.5 \pm 3.2$ | $41.6 \pm 7.9$ | $50.7 \pm 70.4$ | $25.9 \pm 3.6$ | $60.6 \pm 15.6$ | $134.2 \pm 217.1$ |
| $\sqrt{2}$ | $9.6 \pm 0.4$ | $10.8 \pm 0.6$ | $11.7 \pm 1.2$ | $9.2 \pm 0.5$ | $10.5 \pm 0.3$ | $11.3 \pm 0.5$ |

Table 2: The difference is the predicted max clique size under size generalization domain shift. The train domain graphs were constructed by drawing $n \in [40, 50]$ points uniformly in the unit square, and connecting two point if their distance is less than $\rho_{train} = 0.3$. The test set domain graphs contain $n = 100$ nodes, effectively increasing their density by 2. We tested with two different values of $\rho_{train}/\rho_{test}$, the ratio between the train and test connectivity radius. A proper scaling that keeps the expected degree of each node is $\rho = \sqrt{2}$. Here, although proper scaling does not help solve the problem completely, it does improve performance.

| | | Node regression | | Graph regression | |
|---|---|---|---|---|---|
| | $p$ | Student - Teacher | Degree | Student - Teacher | Edge count |
| 1-GNN | 0.3 | $187 \pm 392$ | $11.9 \pm 19.4$ | $60.4 \pm 91.8$ | $(1.2 \pm 1.6) \cdot 10^3$ |
| | 0.15 | $(3.7 \pm 5.1) \cdot 10^{-2}$ | $(7.2 \pm 3.7) \cdot 10^{-5}$ | $(9 \pm 10) \cdot 10^{-4}$ | $(6.9 \pm 3.3) \cdot 10^{-3}$ |
| GIN | 0.3 | $125 \pm 370$ | $2.57 \pm 2.88$ | $33.8 \pm 84.9$ | $284 \pm 286$ |
| | 0.15 | $1.35 \pm 4.5$ | $(2.2 \pm 0.3) \cdot 10^{-2}$ | $(5 \pm 17) \cdot 10^{-2}$ | $(9.1 \pm 7.1) \cdot 10^{-3}$ |

Table 3: Comparing performance on different local distributions (a) A student-teacher graph regression task; (b) A graph regression task, where the graph label is the number of edges; (c) A student-teacher node regression task; (d) A node regression task, where the node label is its degree. In the edge count/degree tasks the loss is the mean difference from the ground-truths, divided by the average degree/number of edges. In the student-teacher tasks the loss is the mean $L_2$ loss between the teacher's value and the student's prediction, divided by the averaged student's prediction. Both the student and teacher share the same 3-layer architecture

Next, we tested on the student-teacher task, on both graph and node levels, on the degree prediction task of each node, and the task of predicting the number of edges in the graph. The goal of these experiments is to show that the size generalization problem persists on different tasks, different architectures, and its intensity is increased for deeper GNN. In all the experiments we draw the graphs from $G(n, p)$ distribution, wherein the test set $n$ is drawn uniformly between 40 and 50, and $p = 0.3$, and in the test set $n = 100$ and $p$ is either 0.3 or 0.15. We note that when $p = 0.15$, the average degree of the test graph is equal to (approximately) the average degree of the train graph, while when $p = 0.3$ it is twice as large. We would expect that the model will generalize better when the average degree in the train and test set is similar, because then their $d$-patterns will also be more similar.

Table 3 compares the performance of these tasks when changing the graph size of the test data. We tested the performance with normalized test loss, where we normalized the output by the average output of the test set. This metric allows us to estimate the percentage of mistakes from the average output.

## D  SSL TASK ON WL TREE

First, we will need the following definition which constructs a tree out of the $d$-pattern introduced in the previous section. This tree enables us to extract certain properties for each node which can, later on, be learned using a GNN. This definition is similar to the definition of "unrolled tree" from (24).

**Definition D.1** ($d$-pattern tree). *Let $G = (V, E)$ a graph, $C$ a finite set of node features, where each $v \in V$ have a corresponding feature $c_v$, and $d \geq 0$. For a node $v \in V$, its d-**pattern tree** $T_v^{(d)} = (V_v^{(d)}, E_v^{(d)})$ is directed tree where each node corresponds to some node in G. It is defined recursively in the following way: For $d = 0$, $V_v^{(0)} = u_{(0,v)}$, and $E_v^{(0)} = \varnothing$. Suppose we defined $T_v^{(d-1)}$, and let $\tilde{V}_v^{(d-1)}$ be all the leaf nodes in $V_v^{(d-1)}$ (i.e. nodes without incoming edges). We define:*

$$V_v^{(d)} = V_v^{(d-1)} \cup \left\{ u_{(d,v')} : v' \in \mathcal{N}(v''), \ u'_{(d-1,v'')} \in \tilde{V}_v^{(d-1)} \right\}$$

$$E_v^{(d)} = E_v^{(d-1)} \cup \left\{ (u_{(d,v')}, u'_{(d-1,v'')}) : v' \in \mathcal{N}(v''), \ u'_{(d-1,v'')} \in \tilde{V}_v^{(d-1)} \right\}$$

*and for every node $u_{(d,v')} \in V_v^{(d)}$, its node feature is $c_{v'}$ - the node feature of $v'$*

The main advantage of pattern trees is that they encode all the information that a GNN can produce for a given node by running the same GNN on the pattern tree.

This tree corresponds to the local patterns in the following way: the $d$-pattern tree of a node can be seen as a multiset of the children of the root, and each child is a multiset of its children, etc. This completely describes the $d$-pattern of a node. In other words, there is a one-to-one correspondence between $d$-patterns and pattern trees of depth $d$. Thus, a GNN that successfully represents the pattern trees of the target distribution will also successfully represent the $d$-patterns of the target distribution.

Using the $d$-pattern tree we construct a simple regression SSL task where its goal is for each node to count the number of nodes in each layer of its $d$-pattern tree. This is a simple descriptor of the tree, which although loses some information about connectivity, does hold information about the structure of the layers.

For example, in Fig. 3 the descriptor for the tree would be that the root (zero) layer has a single black node, the first layer has two yellow nodes, the second layer has two yellow, two gray, and two black nodes, and the third layer has ten yellow, two black and two gray nodes.

# E  TRAINING PROCEDURE

In this section we explain in details the training procedure used in the experiments of Sec. 4. Let $X_{Main}$ and $X_{SSL}$ be two labeled datasets, the first contains the labeled examples for the main task from the source distribution, and the second contains examples labeled by the SSL task from both the source and target distributions. Let $\ell : \mathbb{R} \times \mathbb{R} \to \mathbb{R}$ be a loss function, in all our experiments we use cross entropy loss for classification tasks and squared loss for regression tasks. We construct the following models:

(1) $f_{GNN}$ is a GNN feature extractor. Its input is a graph and its output is a feature vector for each node in the graph. (2) $h_{Main}$ is a head (a small neural network) for the main task. Its inputs are the node feature and it outputs a prediction (for graph prediction tasks this head contains a global pooling layer). (3) $h_{SSL}$ is the head for the SSL task. Its inputs are node features, and it outputs a prediction for each node of the graph, depending on the specific SSL used.

**Pretraining.**  Here, there are two phases for the learning procedure. In the first phase, at each iteration we sample a batch $(\mathbf{x}_1, \mathbf{y}_1)$ from $X_{SSL}$, and train ny minimizing the objective: $\ell(h_{SSL} \circ f_{GNN}(\mathbf{x}_1), \mathbf{y}_1)$. In this phase both $h_{SSL}$ and $f_{GNN}$ are trained. In the second phase, at each iteration we sample $(\mathbf{x}_2, \mathbf{y}_2)$ from $X_{main}$ and train on the loss $\ell(h_{Main} \circ f_{GNN}(\mathbf{x}_2), \mathbf{y}_2)$, where we only train the head $h_{Main}$, while the weights of $f_{GNN}$ are fixed.

**Multitask training.** Here we train all the functions at the same time. At each iteration we sample a batch $(\mathbf{x}_1, \mathbf{y}_1)$ from $X_{SSL}$ and a batch $(\mathbf{x}_2, \mathbf{y}_2)$ from $X_{Main}$ and train by minimizing the objective:

$$\alpha \ell(h_{SSL} \circ f_{GNN}(\mathbf{x}_1), \mathbf{y}_1) + (1 - \alpha)\ell(h_{Main} \circ f_{GNN}(\mathbf{x}_2), \mathbf{y}_2).$$

Here $\alpha \in [0, 1]$ is the weight for the SSL task, in all our experiments we used $\alpha = 1/2$.

For an illustration of the training procedures see Fig. 5. These procedures are common practices for training with SSL tasks (see e.g. (39)).

We additionally use a semi-supervised setup in which we are given a dataset $X_{FS}$ of few-shot examples from the target distribution with their correct label. In both training procedures, at each iteration we sample a batch $(\mathbf{x}_3, \mathbf{y}_3)$ from $X_{FS}$ and add a loss term $\beta\ell(h_{Main} \circ f_{GNN}(\mathbf{x}_3), \mathbf{y})$ where $\beta \in [0, 1]$ is the weight of the few-shot loss. In pretraining this term is only added to the

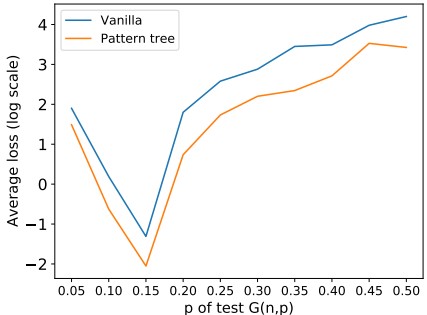

Figure 5: Two training procedures for learning with SSL tasks. **Left:** Learning with pretraining: Here, a GNN is trained on the SSL task with a specific SSL head. After training, the weights of the GNN are fixed, and only the main head is trained on the main task. **Right:** Multitask learning: Here, there is a shared GNN and two separate heads, one for the SSL task and one for the main task. The GNN and both heads are trained in simultaneously.

Figure 6: Teacher-student setup with a 3-layer GNN. Training is on graphs drawn i.i.d from $G(n, p)$ with $n \in \{40, ..., 50\}$ uniformly and $p = 0.3$. Testing is done on graphs with $n = 100$ and $p$ vary (x-axis). The "Pattern tree" plot represent training with our pattern tree SSL task, using the pretraining setup.

second phase, with weight $1/2$ and adjust the weight of the main task to $1/2$ as well (equal weight to the main task). In the multitask setup, we add this term with a weight $1/3$ and adjust the weights of the two other losses to $1/3$ as well, so all the losses have the same weight.

## F    MORE EXPERIMENTS FROM SEC. 4

For synthetic datasets, we used the setting of Section 3.4. Source graphs were generated with $G(n, p)$ with $n$ sampled uniformly in $[40, 50]$ and $p = 0.3$. Target graphs were sampled from $G(n, p)$ with $n = 100$ and $p = 0.3$.

Table 4 depicts the results of using the $d$-patterns SSL tasks, in addition to using the semi-supervised setting. It can be seen that adding the $d$-patterns SSL task significantly improves the performance on the teacher-student task, although it does not completely solve it. We also observe that adding labeled examples from the target domain significantly improves the performance of all tasks. Note that adding even a single example significantly improves performance. In all the experiments, the network was successful at learning the task on the source domain with less than $0.15$ averaged error, and in most cases much less.

Fig. 6 depicts a side-by-side plot of the 3-layer case of Fig. 2 (left), where training is done on graphs sampled from $G(n, p)$ with $40$ to $50$ nodes and $p = 0.3$, and testing on graphs with $100$ nodes and $p$ varies. We compare Vanilla training, and our pattern tree SSL task with pretraining. It is clear that for all values of $p$ our SSL task improves over vanilla training.

### F.1    DATASETS STATISTICS

Table F.1 shows the statistics of the datasets that were used in the paper. In particular the table presents the split that was used in the experiments, we trained on graphs with sizes smaller or equal to the 50-th percentile and tested on graphs with sizes larger or equal to the 90-th percentile. In all

| TASKS | # TARGET SAMPLES | EDGE COUNT | DEGREE | TEACHER STUDENT | TEACHER STUDENT PER NODE |
|---|---|---|---|---|---|
| **VANILLA** | 0 | $1318 \pm 1398$ | $12.22 \pm 15.35$ | $43.1 \pm 38.1$ | $30.47 \pm 64.85$ |
| | 1 | $0.46 \pm 0.68$ | $0.01 \pm 0.01$ | $0.13 \pm 0.22$ | $0.06 \pm 0.15$ |
| | 5 | $0.06 \pm 0.07$ | $<$1E-3 | $0.08 \pm 0.28$ | $0.02 \pm 0.05$ |
| | 10 | $0.03 \pm 0.03$ | $<$1E-3 | $0.01 \pm 0.01$ | $0.01 \pm 0.03$ |
| $d$-**PATTERN PT** | 0 | $1293 \pm 918$ | $53.24 \pm 60.26$ | $2.6 \pm 3.96$ | $0.39 \pm 0.9$ |
| | 1 | $1.7 \pm 1.05$ | $0.07 \pm 0.04$ | $<$1E-3 | $<$1E-4 |
| | 5 | $0.09 \pm 0.19$ | $0.02 \pm 0.005$ | $<$1E-3 | $<$1E-4 |
| | 10 | $0.05 \pm 0.05$ | $0.02 \pm 0.01$ | $<$1E-3 | $<$1E-4 |

Table 4: Test results est of synthetic datasets. Values are the loss divided by the mean over the test set of target output.

Table 5: Dataset statistics.

| | NCI1 | | | NCI109 | | | DD | | |
|---|---|---|---|---|---|---|---|---|---|
| | all | Smallest 50% | Largest 10% | all | Smallest 50% | Largest 10% | all | Smallest 50% | Largest 10% |
| Class A | 49.95% | 62.30% | 19.17% | 49.62% | 62.04% | 21.37% | 58.65% | 35.47% | 79.66% |
| Class B | 50.04% | 37.69% | 80.82% | 50.37% | 37.95% | 78.62% | 41.34% | 64.52% | 20.33% |
| Num of graphs | 4110 | 2157 | 412 | 4127 | 2079 | 421 | 1178 | 592 | 118 |
| Avg graph size | 29 | 20 | 61 | 29 | 20 | 61 | 284 | 144 | 746 |

| | Twitch_egos | | | Deezer_egos | | | IMDB-binary | | |
|---|---|---|---|---|---|---|---|---|---|
| | all | Smallest 50% | Largest 10% | all | Smallest 50% | Largest 10% | all | Smallest 50% | Largest 10% |
| Class A | 46.23% | 39.05% | 58.07% | 56.80% | 44.78% | 64.97% | 50.00% | 48.98% | 55.55% |
| Class B | 53.76% | 60.94% | 41.92% | 43.19% | 55.21% | 35.02% | 50.00% | 51.01% | 44.44% |
| Num of graphs | 127094 | 65016 | 14746 | 9629 | 4894 | 968 | 1000 | 543 | 108 |
| Avg graph size | 29 | 20 | 48 | 23 | 13 | 68 | 19 | 13 | 41 |

| | PROTEINS | | |
|---|---|---|---|
| | all | Smallest 50% | Largest 10% |
| Class A | 59.56% | 41.97% | 90.17% |
| Class B | 40.43% | 58.02% | 9.82% |
| Num of graphs | 1113 | 567 | 112 |
| Avg graph size | 39 | 15 | 138 |

the datasets there is a significant difference between the graph sizes in the train and test sets, and in some datasets there is also a difference between the distribution of the output class in the small and large graphs.

## F.2 DEGREE DISCREPANCY CORRESPONDS TO SIZE DISCREPANCY

In this subsection we calculated the degree histogram of two of the real datasets that we tested on: IMDB-Binary and D & D. For the degree histogram see Fig. 7. Recall the the degree is the 1-pattern of a node. It is clear from the plots that for the IMDB dataset there is a difference in the distribution of degrees between small and large graphs, while in D & D the distributions look almost the same. Correspondingly, In our experiments the pattern tree SSL task significantly improved performance on the IMDB dataset, while not improving performance on the D& D dataset. This gives further evidence that a discrepancy between the $d$-patterns leads to bad generalization, and that correctly representing the patterns of the test set can improve performance. While in this simple experiment we only considered 1-patterns it is worth mentioning that the discrepancy is can only be accentuated when considering deeper patterns.

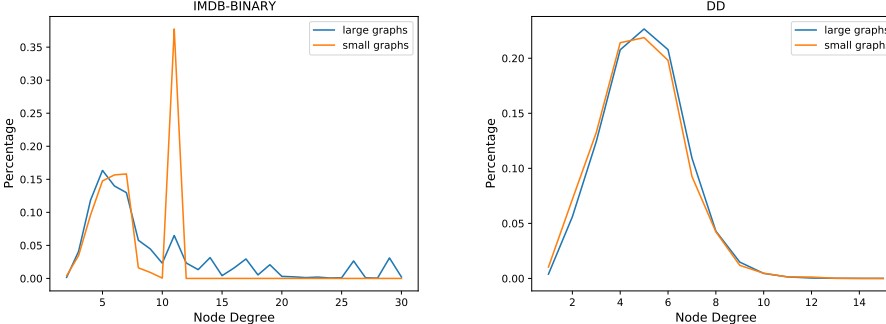

Figure 7: Histogram in percentage of degrees of graphs. We used the 10% smallest and largest graphs in each dataset.

