# OpenReview forum: "On Size Generalization in Graph Neural Networks"
_ICLR.cc/2021/Conference — Reject_

### Official Review · AnonReviewer3 · 2020-10-27
**Is the train and test d-pattern distributions being "close" to each other (e.g., in total-variation sense) a necessary condition for generalization?**

**Rating:** 5
**Confidence:** 4

**Review:**

The paper considers the important problem of lack of generalization of GNNs to graphs whose sizes are much larger than the graphs on which they were trained on. The authors propose a theoretical explanation by arguing that this lack of generalization is due to a mismatch in the “d-patterns” (essentially the pattern of neighborhood around nodes) between the training and test graphs. To improve generalization performance, they borrow ideas from domain adaptation and propose a pattern-tree based pretext learning task which improves classification accuracy compared to baseline pretext tasks in a range of datasets and tasks.

Understanding and improving size generalization in GNNs is an important and timely topic. I appreciate the authors considering this problem and their efforts towards solving it. Overall the paper is well written, and I enjoyed reading it.

While I do agree that a mismatch in the d-pattern distribution can lead to poor generalization (Corollary 3.3 and Equation 1), there are a few aspects to this claim that I do not understand. For instance, suppose my training set consists of graphs whose avg. degrees are 2, 3, …, 100. And suppose the test set consists of only graphs whose avg. degree is 50. Clearly there is a huge mismatch in the d-patterns between the training and test sets (in the sense of Equation 1 in the paper). However intuitively I would expect the GNN to do well on the test set since it would have encountered degree 50 graphs while training. Could you explain this apparent inconsistency?

Following Corollary 3.3., the proposed solutions (d-pattern pretext training or semi-supervised learning) attempt to bring the training pattern distributions closer to the test distribution. This makes sense because if the training distribution got closer to the test distribution, the generalization would be better. However, in the case of SSL adding only a few labelled examples (1, 5 or 10) from the target domain already provides a sizeable improvement in performance. Similarly, in Appendix E the authors report that “adding just a single labelled example from the target distribution significantly improves performance”. Intuitively adding a small number of examples from the target domain should not change the training distribution by much, and yet the performance gains are substantial. I again cannot see how Corollary 3.3 can explain this phenomenon.

In the proof of Corollary 3.3 (in Appendix B), it states that “by thm 3.2 there exists a 1-GNN such that for any d-pattern p_i \in A gives a wrong label and for every pattern outside A, p_j \in A^c gives the correct label”. As far as I understand, thm 3.2 says there exists a GNN that can predict the labels correctly for every p_j \in A^c. I do not follow how thm 3.2 says that for patterns outside of A^c the GNN would predict the labels incorrectly. Could you please clarify?

In Figure 1, why do GNNs with more layers tend to do worse compared to GNNs with just one layer? It would be good to have a similar plot where you use your proposed DA solution. While you do have some results in Appendix E, having a side-by-side comparing before and after pretraining or SSL would help highlight the effectiveness of the solution better. Have you tried regularization to improve generalization?

In Figure 3, could you explain the trend when the number of labelled large graph examples is more than 10?

---

> ### Author Response · Authors · 2020-11-17
> **Answer to Reviewer 3**
>
> We thank the reviewer for the thorough review and for the very insightful comments. We are happy that you enjoyed reading the paper.
>
> Q: “Is the train and test d-pattern distributions being "close" to each other ... a necessary condition for generalization?“
> A: Our theoretical result says that in case the d-pattern distributions are not close to each other in the total-variation sense, there exists a bad solution that does not generalize. On the other hand, in theory, there might be other “good” solutions that do generalize depending on the task at hand. The theoretical question of convergence to bad or good local minima is still not solved.
> We do, however, study this question empirically in Subsection 3.4, where we show that generalization is inversely correlated to total variation distance, i.e., small total variation distance yields size generalization solutions.
>
> Q: “... suppose my training set consists of graphs whose avg. degrees are 2, 3, …, 100. And suppose the test set consists of only graphs whose avg. degree is 50. ... intuitively I would expect the GNN to do well on the test set ... Could you explain this apparent inconsistency?”
> A: Thank you for this great example. If we assume a uniform distribution over degrees in this example, our theoretical results assure that there exists a bad solution that will not generalize in that case. Our empirical results, however,  show a slightly more intricate result:  in case there are d-patterns that appear in the test but not appear on the train, then training will lead to those non-generalizing solutions. In your example, the test distribution is represented in the train distribution and the network might not overfit.  We made it clearer in the revision. It is an interesting future question to study the tradeoff between the total-variation distance, and the generalization capabilities of GNNs.
>
> Q: “... in the case of SSL adding only a few labeled examples...already provides a sizable improvement in performance...”
> A: We give an equal weight for the few labeled examples from the target domain and the entire distribution of the source domain. This is explained in the “Architecture and training protocol” paragraph at the beginning of Subsection 4.1. This is a standard approach for applying semi-supervised learning and it indeed changes the training data distribution significantly, thus it actually supports the claim of the Corollary. Also note that for each graph in the few labeled examples there are many d-patterns, as many as the number of nodes.
>
> Q: In the proof of Corollary 3.3 … I do not follow how thm 3.2 says that for patterns outside of A^c the GNN would predict the labels incorrectly...”
> A: We use the assumption of Corollary 3.3 that the graph distributions have finite support, meaning that there are only a finite number of graphs. Thus, there is a finite number of d-patterns in both A and A^c, and now we can use Thm. 3.2. We clarified this in the proof of the revised version. The finite support assumption is natural since many real graph learning problems are defined over a finite set of categorical features over graphs with a bounded number of nodes.
>
> Q: “In Figure 1, why do GNNs with more layers tend to do worse compared to GNNs with just one layer?“
> A: Indeed this is the trend we saw in all of our experiments. We can offer the following explanation: Theorem 3.2 has specific conditions on the depth of the networks so they can overfit a certain set of d-patterns: the depth should be d+2. Adding more layers brings us closer to the settings of this theorem and thus enables the network to perform a stronger overfit, over deeper d-patterns (i.e. larger d).
>
> Q: “It would be good to have a similar plot where you use your proposed DA solution...“
> A: We added a side-by-side plot comparing the same experiments made in Section 3 with or without our pretraining SSL task. Please see Figure 6 in the appendix. The results show consistent improvement when using our suggested SSL task.
>
> Q: “Have you tried regularization to improve generalization?”
> A: We performed extensive experiments, including adding regularization terms, specifically L_1 and L_2 regularization, which didn’t help in size generalization. For example, in our Vanilla setup (Section 4) we include an L_2 regularization term.  We also showed in the toy example (Subsection 3.1) that even in very simple tasks, regularization does not solve the problem.
>
> Q: “In Figure 3, could you explain the trend when the number of labeled large graph examples is more than 10?”
> A: The effect of extra labeled examples saturates at ~10 examples. This is because as the number of few-shot examples grows, there is less size discrepancy between the train and test sets, and there is no need for an SSL task. We performed an extended experiment and saw that, as expected, as more few shot examples are added, vanilla training and our SSL task tend to have the same performance.

---

### Official Review · AnonReviewer4 · 2020-10-28
**Novel and well-motivated investigation into GNN size generalization error**

**Rating:** 7
**Confidence:** 3

**Review:**

Summary:
This paper investigates the issue of generalizability of GNNs when trained on small graphs and tested on larger graphs, a common setting in graph learning. The paper argues that the ability of constant-depth GNNs to generalize is not dependent on the size difference, but on the difference in distributions of nodes' neighborhood features, which authors call "d-patterns." The paper shows theoretically that a GNN can be trained to achieve zero loss on small graphs, yet fail to generalize to larger graphs and the loss on these graphs will be dependent on the difference in d-pattern distributions.

Strengths:
* Shows strong theoretical and empirical evidence of GNNs inability to generalize to graphs larger than those seen in training when the distribution of d-patterns of the graphs differ, showing that techniques like regularization will not help in these cases
* Provides a first step to mitigate problem by using domain adaptation methods and in particular using an intermediate task of identifying the d-patterns in each graph.

Weaknesses:
* The work only considers constant-depth GNNs and local tasks
* Generalization error is shown only for 0-1 loss, limiting the graph problems that it applies to to node live binary classification and functions thereof.

Conclusion:
The failures of GNNs to generalize to larger graphs in many cases is a known phenomenon. This paper's investigation of this issue and a clear explanation of at least one reason for it is clear and well supported. Though a more minor contribution to the paper, the proposal for some mitigating techniques (DA and in particular the pattern tree labeling task) can serve as motivation for further investigation of models robust to these generalization errors. I recommend acceptance based on the clear motivation and development of the investigation presented in the work.

update: Thank you to authors for their response to reviewer comments. I acknowledge I have read and reviewed their rebuttals.

Questions:
1. The claim made is that the distribution shift in d-patterns is the driving factor in generalization error. The experiments in 3.4 support this, but one experiment similar to those shown in Figure 1 that would illustrate this further is training on sizes n=[40,x] and p=0.3 as in Figure 1 (right) and testing on n=150 and p=0.002*[40,x]. If the size difference between train and test is not the driving factor, but rather d-patters, should we expect such an experiment would yield a mostly-flat loss curve? Would the value of the loss on the test graphs be similar to the loss achieved on graphs from the original distribution (n=[40, x] with p=0.03)?
2. The focus of the work is on size generalization and in particular small-to-large graph domain shift. The claims appear to also hold for evaluating generalization on similarly sized graphs with differences in node attributes and/or connectivity patterns. Is this the case, why or why not?

Minor comments and typos:
*  In the definition of d-patterns, the integer $\ell$ is not defined. Is this the number of neighbors, or the number of possible d-patterns?
* Top of page 8 “does have node features” should be “does *not* have node features”
* Page 8, “Datasets” paragraph says “and the and 10% largest graphs.” I assume this should read “and the largest 10% of graphs…”
* “The” should be capitalized in the last paragraph (“Results”) at the beginning of the sentence “The accuracy monotonically increases…”

---

> ### Author Response · Authors · 2020-11-17
> **Answer to Reviewer 4**
>
> We are very thankful for your useful remarks and questions.
>
> Q: “The work only considers constant-depth GNNs and local tasks”
> A: GNNs with fixed depth is by far the most popular architecture used today. We focus on this case because it is the most relevant setup to the community. Currently, there are only a handful of models using variable depth.
>
> Q: “Generalization error is shown only for 0-1 loss, limiting the graph problems that it applies to node live binary classification and functions thereof.”
> A: We focus on 0-1 loss for the terseness of the proofs. Our claims can be readily extended to other bounded losses. We stress that we also considered graph classification in our paper. In the revision, we significantly extended our treatment of this case.
>
> Q: “The claim made is that the distribution shift in d-patterns is the driving factor in generalization error. The experiments in 3.4 support this, but one experiment similar to those shown …”
> A: Your claim is correct, we added an experiment in the spirit of your suggestion to the revised version. In the experiment, we train on sizes [40,50] and p=0.3 and test on growing sizes but normalizing the p parameter so that the mean degree stays roughly the same in expectation. The plot is indeed almost flat. Please see Figure 2 (right).
>
> Q: “The focus of the work is on size generalization and in particular small-to-large graph domain shift. The claims appear to also hold for evaluating generalization on similarly sized graphs with differences in node attributes and/or connectivity patterns. Is this the case, why or why not?”
> A: You are correct. Our theoretical results (specifically Corollary 3.4) hold for any two graph distributions for which the d-patterns have a large total variation distance. We phrased the theorems as they are because we wanted our work to focus on size generalization problems, which we think are one of the most important setups in which such domain shifts appear.
> We also thank the reviewer for the minor comments and typos, we fixed and clarified them in the revised version of the paper.

---

### Official Review · AnonReviewer1 · 2020-10-28

**Rating:** 4
**Confidence:** 5

**Review:**

This paper proposes a notion of ''d-patterns'' for graphs -- 1-pattern corresponds to the degree of each node; 2-pattern is for each possible degree i the number of neighbors with degree i; 3-pattern is for each possible 2-pattern, the number of its neighbors with the exact 2-pattern; N-pattern are defined similarly. The paper shows that given a set of d-patterns, there exists an assignment of GNN weight that fits this d-pattern output, so there exist some assignments of weights to GNNs that cannot generalize to larger graphs. Based on the existence of a set of bad parameters, authors claim that GNNs will fail to size generalize. Authors also claim that d-pattern is the determine factor of whether a GNN can size generalize. In the last two pages, the paper proposes to improve size generalization ability of GNNs by self-supervised training on unlabelled/labeled data with a d-pattern / pattern-tree task. Some experiments are conducted to demonstrate pattern-tree task slightly improves vanilla GNNs.

This paper is well-written and easy to follow.  There are several major concerns regarding the significance and correctness of the results.

a) In the main result Corollary 3.3, authors claim that because there exists a weight assignment of GNN that cannot generalize to lager graphs given discrepancy on d-pattern, then GNN will fail to generalize to larger graphs given d-pattern discrepancy. Authors also claim d-pattern should be the determining factor for size generalization.

This main result/claim has several problems.

- The existence of a bad set of weight parameters does not imply training GNN will necessarily converge to this parameter. Hence you cannot use this to claim GNN will fail to size generalize.
- I am not convinced d-pattern is the determining factor for size generalization. The d-pattern definition is quite complicated and it is not necessarily an important factor in many tasks. In fact, your argument of the existence of a set of bad parameters can be easily constructed with other measures, not just d-pattern.
- To claim d-pattern is the main factor, you need to prove that when there is no d-pattern discrepancy, GNN will always size generalize. Moreover, you need to prove when there is d-pattern discrepancy, GNNs will definitely fail (note that the existence proof is not valid for this purpose).

b) Authors claim one of their main contributions is to find out size generalization is difficult. But size generalization is hard is already well-known and not surprising. Many previous works have demonstrated this.

c) The proposed method (self-supervised training with pattern tree task) is incremental. Self-supervised training of GNNs is not new, the fact that pre-training can improve robustness to larger graphs is also not new and have been shown in previous works. The pattern tree task is not significant either.

I encourage authors to propose a more novel method and thoroughly demonstrate its effectiveness before submission to the next venue.


Some references that demonstrate size generalization is a well-known issue.

References

Measuring abstract reasoning in neural networks. Barrett et al 2018.

Learning TSP Requires Rethinking Generalization. Joshi et al 2020.

Learning Combinatorial Optimization Algorithms over Graphs. Dai et al 2017.

---

> ### Author Response · Authors · 2020-11-17
> **Answer to Reviewer 1**
>
> We thank the reviewer for the very useful remarks, and happy that you found our paper well written and easy to follow.
>
> Q: “I am not convinced d-pattern is the determining factor for size generalization…”
> A: This is actually a very natural and popular notion for studying GNNs. The definition of d-patterns was used (in a different form) in previous papers that studied the expressivity of GNNs, in particular, in papers that compare the power of GNNs and the Weisfeiler-Lehman algorithm (e.g Morris et al 2019, Xu et al. 2019). Any other descriptor that can capture the expressivity of MPNNs is either weaker than d-patterns (due to the results from our paper and from Morris et al 2019), or cannot be captured by GNNs. For example, the d-hop neighborhood of a node cannot be fully captured by MPNNs (e.g. different neighborhoods may give rise to the same d-patterns). We added a theorem in the revised paper (Thm. 3.2) that better explains the power of d-patterns. Combined with Thm 3.3, it follows that d-patterns are the only factor for determining the expressive power of GNN.
>
> Q: “ ...The existence of a bad set of weight parameters does not imply training GNN will necessarily converge to this parameter. Hence you cannot use this to claim GNN will fail to size generalize.”
> A: As stated before, this is not our argument. Our theoretical results only consider the expressive power of GNNs, we are the first to show that there are bad solutions that will size overfit, which was not known before. Hence, we put forward the size generalization problem theoretically by showing that, at least on the expressive side, it exists.
> Moreover, we demonstrate empirically that in fact learning on small graphs yields solutions that do not generalize to larger graphs. It is an interesting future direction to show *theoretically* that training GNNs with gradient methods will result in such a solution, but we note that this might be difficult. For example, similar results for fully connected networks are only available for specific unrealistic architectures (shallow networks, infinite width, etc.).
>
> Q: “To claim d-pattern is the main factor, you need to prove that when there is no d-pattern discrepancy, GNN will always size generalize.”
> A: This is true, and easily derived from the definition of d-patterns. We added Corollary 3.5 which proves that when there is no discrepancy between the distributions then every solution that succeeds on small graphs will generalize to larger graphs.
>
> Q: “...you need to prove when there is d-pattern discrepancy, GNNs will definitely fail.”
> A: It is an interesting future direction to show *theoretically* that training GNNs with gradient methods will result in such a solution. Here we show theoretically an expressivity result that these non-generalizing solutions exist, then we show empirically that in certain cases, training GNNs leads to such non-generalizing solutions, even when a good generalizing solution exists.
>
> Q: “Authors claim one of their main contributions is to find out size generalization is difficult... Many previous works have demonstrated this.”
> A: As far as we know, our paper is the first work to study the problem of size generalization theoretically and prove that there exist bad non-generalizing solutions. We also show empirically that training GNNs in practice leads to those non-generalizing solutions, even for very simple tasks.
> We stress that many recent empirical papers claim that size generalization can be achieved, which implies that the question of whether size generalization is a real problem is not a well-known fact yet: some examples of recent papers that achieved size generalization are: Learning to Simulate Complex Physics with Graph Networks, Learning Algebraic Multigrid Using Graph Neural Networks, Neural Execution Of Graph Algorithms Neural Bipartite Matching. Hence, we feel that a theoretical and empirical investigation of the reasons for size generalization, as we provide in our paper, can be relevant to the community.
>
> Q: “The proposed method...is incremental. Self-supervised training of GNNs is not new, the fact that pre-training can improve robustness to larger graphs is also not new...”
> A: We certainly don’t claim to introduce self-supervised learning for GNNs and we clearly cite several such previous works.  On the other hand, As far as we know we are the first to show that certain SSL tasks can mitigate the size generalization problem, at least partially.  Recent papers on self-supervised learning on GNNs (e.g., Hu et al. 2019, You et al. 2020) do not focus on size generalization, but rather on different challenges. We compare our pattern tree task to several other widely used SSL tasks and show that it outperforms them significantly when there is a size discrepancy between the train and test sets. Moreover, our proposed SSL task is only part of our work, as we also put forward the size generalization problem, and study it both theoretically and empirically.

---

### Official Review · AnonReviewer2 · 2020-10-28
**Official Blind Review #2**

**Rating:** 5
**Confidence:** 3

**Review:**

This paper theoretically discusses the size generalization problem in GNNs. It introduces the concept of d-patterns and shows how generalization is related to the distribution of d-patterns.

Pros:
+ The size generalization problem in GNN is an important and interesting topic, which has many real-world applications.
+  Though I did not check the proof details, the theoretical results look sound to me.

Cons:
- The proposed method for improving size-generalization is not clearly stated. In Sec 4, it will be better if the authors could clearly write down the training procedure and loss, instead of describing them in words.
- It is also not clear how the proposed method for improving size-generalization is related to the theorem in Sec 3.
- The focus of this paper is on local tasks that can be solved by GNNs with fixed depths, no matter what the graph size is. This is quite a strong assumption, especially on node prediction tasks. Based on this strong assumption, it is clear that the node prediction is only related to its local d-hop subgraph, no matter what the graph size is. Suppose we view each the prediction on each node as a single task, and the input is the d-hop subgraph. Then (1) it is clear that the generalization will depend on the distribution of the d-hop subgraphs only and (2) this problem becomes irrelevant to the size of the original graph. It is then not clear what new information, insight, or conclusion we could draw from the theorem.
- The discussion in this paper does not reveal to be very relevant to the 'size' of the graph. It is not clear how the 'size' plays the role in the generalization. Given two distributions of graphs of the SAME SIZE, the same results will hold, depending on the distributions of the d-patterns.
- In the experiment, it will be better if the authors could show how the performance varies when the graph size becomes larger and larger.

Overall the information and the key message from this paper do not seem to be clear and strong enough. I vote for rejection.

=== after author response ===
I would like to thank the authors for their detailed response. My major concern is the strong assumption in this paper - that the function to be approximated is a fixed-depth GNN. This assumption makes the problem less relevant to the actual size of the graph, and avoids a major challenge in size-generalization - that larger graphs are expected to require deeper operations.

As pointed out in the author's response, some efforts are made to **experimentally** (1) demonstrate the effectiveness of the proposed method on a real dataset which might not be solvable by constant depth GNN, and (2) demonstrate in real datasets there is a discrepancy between the degrees of small and large graphs. I would be happy to raise my score from 4 to 5 given these experiments. However, the theory part could still be further improved so I do not further raise the score.

---

> ### Author Response · Authors · 2020-11-17
> **Answer to Reviewer 2**
>
> We thank the reviewer for the helpful comments and suggestions and are happy the reviewer found the topic of size generalization important and interesting.
>
> Q: “The proposed method for improving size-generalization is not clearly stated.”
> A: Due to space constraints, we described our proposed SSL task in detail in appendix D, including an example and a formal definition of the pattern tree. We also added a thorough explanation, including figures, on the loss and training procedure with SSL in the revised version. Please see Appendix E and Figure 5.
>
> Q: “It is also not clear how the proposed method for improving size-generalization is related to the theorem in Sec 3.”
> A: The main take-home message from our theoretical analysis is that d-patterns that were not seen during training might not be represented well by the network. Consequently, we suggest a method that tries to make sure that off-distribution patterns are represented well by the GNN. The pattern tree is an analogous way to view the d-patterns, each pattern tree uniquely corresponds to a d-pattern. This is explained in detail in appendix D due to space constraints. In the revision, we extended our explanations in the appendix and added a discussion to the main part of the paper. Please see the “Pattern-tree pretext task” paragraph in Section 4.
>
> Q: “The focus of this paper is on local tasks that can be solved by GNNs with fixed depths, no matter what the graph size is.“
> A: GNNs with fixed depth is by far the most popular architecture used today and we focus on this case since it is the most relevant setup for the community. We stress that there is only a handful of papers that use variable depth, and they are designed to handle different problems, such as performing sequential algorithms with GNNs.
> We also note that we focus on tasks solvable by a constant depth GNN *only* in the theoretical part, Importantly, this focus enables us to perform the first theoretical analysis of the problem and show that it exists in very simple settings. Furthermore, in the experimental part, we show that for real datasets (which might not be solvable by constant depth) size generalization is a problem, and our method helps mitigating it.
>
> Q: “it is clear that the node prediction is only related to its local d-hop subgraph, no matter what the graph size is.”
> A: This is correct, but note that our result is also highly relevant for *graph prediction* tasks which strongly depend on the graph size. Specifically, Theorem 3.2 enables us to prove a similar expressivity result for graph prediction tasks. This is discussed in the paper (end of subsection 3.3) and in the revision, we added a formal statement and proof, please see Corollary 3.6.
>
> Q: “The discussion in this paper does not reveal to be very relevant to the 'size' of the graph”
> A: Our paper shows that d-pattern discrepancy is one of the factors that can cause size-overfit. Moreover, we believe that the discussion in this paper is relevant to real applications as in many real-life scenarios, size discrepancy is coupled with d-pattern discrepancy. For example, it is pretty common to see higher node degrees in larger graphs in certain datasets. This is supported by our experiments on real datasets. We also added a subsection in the appendix showing that in real datasets there is a discrepancy between the degrees of small and large graphs. Please see Subsection F.2 and Figure 7.
> Having said that, We agree that the results in this paper can be generalized to other learning setups where there is a discrepancy in the d-patterns (which are not related to the size of the graph) and we believe that leveraging our results for such scenarios is an interesting future research direction. In this paper, we decided to focus on the size generalization problem.
>
> Q: “In the experiment, it will be better if the authors could show how the performance varies when the graph size becomes larger and larger.”
> A: We added another experiment which shows that training on constant range size (40-50) and testing on different sizes (50-200) results in worse results as the graph size increases. Please see Figure 2 (left). Also, Figure 1 (right) studies the effect of increasing the graph sizes in the training while testing on a constant size. It is clear that as the sizes of the train and test are getting closer, the generalization capabilities improve.

---

### Public Comment · ~Benedek_Rozemberczki1 · 2020-11-10
**Attribution of the Deezer and Twitch datasets**

The Deezer and Twitch graph classification datasets were introduced in this paper:

@inproceedings{karateclub,
       title = {{Karate Club: An API Oriented Open-source Python Framework for Unsupervised Learning on Graphs}},
       author = {Benedek Rozemberczki and Oliver Kiss and Rik Sarkar},
       year = {2020},
       pages = {3125–3132},
       booktitle = {Proceedings of the 29th ACM International Conference on Information and Knowledge Management},
       organization = {ACM},
}

Could you add the appropriate citation?

---

> ### Author Response · Authors · 2020-11-17
> **Added reference**
>
> Thanks for pointing this out, we added the reference in the revision.

---

### Author Response · Authors · 2020-11-17
**General Comments + Revision**

We thank the reviewers for their helpful comments and insightful reviews. We are encouraged that reviewers found the size generalization problem in GNNs interesting (R1,R2,R3,R4), the paper is well-written and easy to follow (R1, R3). They also pointed out that we showed strong theoretical and empirical evidence (R4) and recognized that we provide the first steps to mitigate the problem (R4). We stress that to the best of our knowledge, this paper is the first paper to address the important size-generalization problem in graph neural networks theoretically.


We address here the shared concerns, and address individual concerns separately:
1) “The work only considers constant-depth GNNs ”: Our paper focuses on constant-depth GNNs because it is by far the most widely used GNN architecture and the one that the community finds most relevant. Generalizing results to data-dependent depth could be an interesting future question, but, since only a handful of papers discuss this setting, we defer it to future work.


2) “The existence of a bad set of weight parameters does not imply training GNN will necessarily converge to this parameter.”: This is absolutely correct. Our theoretical study proves that there exist weight assignments that size-overfit. That was not known before. Our empirical results show that size-overfitting happens frequently in practice. We emphasized this point in the revised version, in the introduction. We do not claim to show *theoretically* that training GNNs will lead to those solutions, although this is an interesting question for future research.



3) “The results of the paper also hold when there is a discrepancy of d-patterns, rather than a discrepancy between graph sizes”: Thank you for the suggestion. We agree that the results on d-patterns can be used more generally than for size generalization. This is discussed in section 5. More specifically, the theoretical results in the paper are phrased to directly address the size generalization problem since this is the focus of the paper. These results can also be used to address other learning setups where there is a discrepancy of d-patterns between the training and test sets. We believe that this is an advantage of our analysis.

We uploaded a revised version addressing the reviewer's questions and concerns with the following additions. All the revisions from the original paper are marked in blue.
1) Thm. 3.2 and Corollary 3.5 which states that d-layer GNNs are constant on d-patterns (that is, the output of d-layer GNNs on nodes with the same d-pattern is always the same )and that if there is no discrepancy between the d-patterns, every GNN that succeeds on small graphs also generalizes to larger graphs.
2) Corollary 3.6, which gives a generalization result for graph-level tasks (generalizing our result on node-level tasks)
3) Added experiments depicted in Figure 2 - training on constant graph sizes and testing and varying sizes. Two cases are considered - constant p in the G(n,p) distribution of the test set or normalized p. It can be clearly seen that when p is normalized then the discrepancy of d-patterns is small and GNNs successfully generalize, while for constant p they don't.
4) Figure 6 in the appendix - a side-by-side comparison between the pattern tree SSL task (with pretraining) and vanilla training. The SSL task consistently improves the results.
5) Appendix E and Figure 5 - a thorough description of the training procedures (pretraining and multitask), the objective in each case, and figures that illustrate the architectures.
6) We added more explanations to Appendix D - explaining in detail the pattern tree SSL task and connecting it to the d-patterns. The main point is that there is a one-to-one correspondence between pattern trees and d-patterns, so learning patterns trees encourages a good representation of the corresponding d-patterns. We also added a short explanation in the main part of the paper (Section 4)
7) Appendix F.2 and Figure 7 - the distribution of the degree (1-patterns) in two of the real datasets that were tested. The dataset that benefits from the tree pattern SSL task have a high discrepancy of degree distribution between small and large graphs, and thus also a high discrepancy of d-pattern distribution, while the dataset with a mild discrepancy indeed does not benefit much from our SSL task.

---

> ### Comment · AnonReviewer1 · 2020-11-17
> **Not convinced by author response to Concern #2**
>
> This is regarding author response to Concern 2 which I raised in my review.
>
> “The existence of a bad set of weight parameters does not imply training GNN will necessarily converge to this parameter.” This is absolutely correct. Our theoretical study proves that there exist weight assignments that size-overfit. That was not known before. Our empirical results show that size-overfitting happens frequently in practice. We emphasized this point in the revised version, in the introduction. We do not claim to show theoretically that training GNNs will lead to those solutions, although this is an interesting question for future research.
>
> I thank authors for the detailed response, however, I do not think the response is a strong one and I am not convinced.
>
> 1. There is still a logical gap between what the authors claim in the title, abstract, introduction and story and what the authors actually manage to prove: the existence of bad minima doesn't imply bad generalization. In fact, there are many (both classic and modern) results showing good generalization and convergence to global minima despite many poor minima. This logical gap is crucial and misleading for our understanding of GNN size generalization.
>
> Regarding the authors' actual contribution, if they rephrase the paper (title, abstract, intro) as "There exist bad parameters for GNN", it would be more informative and accurate. The current results perhaps do not warrant the title and story "on size generalization of GNN".
>
> 2. A large body of prior literature has already demonstrated the fact that size generalization is hard, as I have mentioned in my review. It is unfair to these works for this paper to ignore them without proper citation, and claim the empirical contribution as its current form.

---

> > ### Author Response · Authors · 2020-11-22
> > **Response to Reviewer #1**
> >
> > We thank the reviewer for the response.
> >
> > (1) We now further clarified which of the claims in the abstract and introduction are based on proofs, and which on experiments.
> >
> > (2) We made a good faith effort to refer to all prior literature that is relevant. We would be grateful to the reviewer for pointing to specific missing references.

---

### Decision · Program_Chairs · 2021-01-07
**Final Decision**

**Decision:**

Reject

**Comment:**

The paper seeks to point out the difficulty of size generalization in GNNs for node prediction and analyze why this happens. The analysis is anchored on the construction of so called d-patterns. The main argument is presented in Corollary 3.4 which shows that discrepancy in d-patters between small and large graphs introduces the possibility of finding a GNN that fits the d-patterns well in a small graph while yielding poor answers on a large graph (assuming the task is solvable by a constant depth GNN). While good to show, the existence of bad parameters does not necessarily suggest that this setting is likely to arise after training. As pointed out by AnonReviewer1, size generalization is also not a new issue but has already been introduced / highlighted in previous publications (Barrett et al 2018, Joshi et al 2020, Dai et al 2017). The authors do provide an algorithm and empirical arguments to mitigate the effect of d-pattern discrepancy. A revised version of the manuscript can hopefully make a stronger case for d-patterns, constant depth computation, and the relevance of these for size generalization.